# Handling correlated and repeated measurements with the smoothed multivariate square-root Lasso

**Quentin Bertrand** *
Université Paris Saclay, Inria, CEA
Palaiseau, 91120, France
quentin.bertrand@inria.fr

**Mathurin Massias** *
Université Paris Saclay, Inria, CEA
Palaiseau, 91120, France
mathurin.massias@inria.fr

**Alexandre Gramfort**
Université Paris Saclay, Inria, CEA
Palaiseau, 91120, France
alexandre.gramfort@inria.fr

**Joseph Salmon**
Univ. Montpellier, CNRS
Montpellier, France
joseph.salmon@umontpellier.fr

## Abstract

A limitation of Lasso-type estimators is that the optimal regularization parameter depends on the unknown noise level. Estimators such as the concomitant Lasso address this dependence by jointly estimating the noise level and the regression coefficients. Additionally, in many applications, the data is obtained by averaging multiple measurements: this reduces the noise variance, but it dramatically reduces sample sizes and prevents refined noise modeling. In this work, we propose a concomitant estimator that can cope with complex noise structure by using non-averaged measurements, its data-fitting term arising as a smoothing of the nuclear norm. The resulting optimization problem is convex and amenable, thanks to smoothing theory, to state-of-the-art optimization techniques that leverage the sparsity of the solutions. Practical benefits are demonstrated on toy datasets, realistic simulated data and real neuroimaging data.

## 1 Introduction

In many statistical applications, the number of parameters $p$ is much larger than the number of observations $n$. A popular approach to tackle linear regression problems in such scenarios is to consider convex $\ell_1$-type penalties, as popularized by Tibshirani (1996). The use of these penalties relies on a regularization parameter $\lambda$ trading data fidelity versus sparsity. Unfortunately, Bickel et al. (2009) showed that, in the case of white Gaussian noise, the optimal $\lambda$ depends linearly on the standard deviation of the noise – referred to as *noise level*. Because the latter is rarely known in practice, one can jointly estimate the noise level and the regression coefficients, following pioneering work on concomitant estimation (Huber and Dutter, 1974; Huber, 1981). Adaptations to sparse regression (Owen, 2007) have been analyzed under the names of square-root Lasso (Belloni et al., 2011) or scaled Lasso (Sun and Zhang, 2012). Generalizations have been proposed in the multitask setting, the canonical estimator being Multi-Task Lasso (Obozinski et al., 2010).

The latter estimators take their roots in a white Gaussian noise model. However some real-world data (such as magneto-electroencephalographic data) are contaminated with strongly non-white Gaussian noise (Engemann and Gramfort, 2015). From a statistical point of view, the non-uniform noise level case has been widely explored: Daye et al. (2012); Wagener and Dette (2012); Kolar and Sharpnack (2012); Dalalyan et al. (2013). In a more general case, with a correlated Gaussian noise

---

model, estimators based on non-convex optimization problems were proposed (Lee and Liu, 2012) and analyzed for sub-Gaussian covariance matrices (Chen and Banerjee, 2017) through the lens of penalized Maximum Likelihood Estimation (MLE). Other estimators (Rothman et al., 2010; Rai et al., 2012) assume that the inverse of the covariance (the *precision matrix*) is sparse, but the underlying optimization problems remain non-convex. A convex approach to regression with correlated noise, the Smooth Generalized Concomitant Lasso (SGCL) was proposed by Massias et al. (2018a). Relying on smoothing techniques (Moreau, 1965; Nesterov, 2005; Beck and Teboulle, 2012), the SGCL jointly estimates the regression coefficients and the noise *co-standard deviation matrix* (the square root of the noise covariance matrix). However, in applications such as M/EEG, the number of parameters in the co-standard deviation matrix ($\approx 10^4$) is typically equal to the number of observations, making it statistically hard to estimate accurately.

In this article we consider applications to M/EEG data in the context of neuroscience. M/EEG data consists in recordings of the electric and magnetic fields at the surface or close to the head. Here we tackle the *source localization* problem, which aims at estimating which regions of the brain are responsible for the observed electro-magnetic signals: this problem can be cast as a multitask high dimensional linear regression (Ndiaye et al., 2015). MEG and EEG data are obtained from heterogeneous types of sensors: magnetometers, gradiometers and electrodes, leading to samples contaminated with different noise distributions, and thus non-white Gaussian noise. Moreover the additive noise in M/EEG data is correlated between sensors and rather strong: the noise variance is commonly even stronger that the signal power. It is thus customary to make several repetitions of the same cognitive experiment, *e.g.,* showing 50 times the same image to a subject in order to record 50 times the electric activity of the visual cortex. The multiple measurements are then classically averaged across the experiment's repetitions in order to increase the signal-to-noise ratio. In other words, popular estimators for M/EEG usually discard the individual observations, and rely on Gaussian i.i.d. noise models (Ou et al., 2009; Gramfort et al., 2013).

In this work we propose Concomitant Lasso with Repetitions (CLaR), an estimator that is

- designed to exploit all available measurements collected during repetitions of experiments,
- defined as the solution of a *convex* minimization problem, handled efficiently by proximal block coordinate descent techniques,
- built thanks to an *explicit* connection with nuclear norm smoothing[1]. This can also be viewed as a partial smoothing of the multivariate square-root Lasso (van de Geer and Stucky, 2016),
- shown (through extensive benchmarks *w.r.t.* existing estimators) to leverage experimental repetitions to improve support identification,
- available as open source code to reproduce all the experiments.

In Section 2, we recall the framework of concomitant estimation, and introduce CLaR. In Section 3, we detail the properties of CLaR, and derive an algorithm to solve it. Finally, Section 4 is dedicated to experimental results.

## 2   Concomitant estimation with correlated noise

**Probabilistic model**   Let $r$ be the number of repetitions of the experiment. The $r$ observation matrices are denoted $Y^{(1)}, \dots, Y^{(r)} \in \mathbb{R}^{n \times q}$ with $n$ the number of sensors/samples and $q$ the number of tasks/time samples. The mean over the repetitions of the observation matrices is written $\bar{Y} = \frac{1}{r} \sum_{l=1}^{r} Y^{(l)}$. Let $X \in \mathbb{R}^{n \times p}$ be the design (or gain) matrix, with $p$ features stored column-wise: $X = [X_{:1} | \dots | X_{:p}]$, where for a matrix $A \in \mathbb{R}^{m \times n}$ its $j^{th}$ column (*resp.* row) is denoted $A_{:j} \in \mathbb{R}^{m \times 1}$ (*resp.* $A_{j:} \in \mathbb{R}^{1 \times n}$). The matrix $B^* \in \mathbb{R}^{p \times q}$ contains the coefficients of the linear regression model. Each measurement (*i.e.,* repetition of the experiment) follows the model:

$$\forall l \in [r], \quad Y^{(l)} = XB^* + S^* E^{(l)} \ , \tag{1}$$

where the entries of $E^{(l)}$ are i.i.d. samples from standard normal distributions, the $E^{(l)}$'s are independent, and $S^* \in \mathcal{S}_{++}^n$ is the co-standard deviation matrix, and $\mathcal{S}_{++}^n$ (*resp.* $\mathcal{S}_+^n$) stands for the set of positive (*resp.* semi-definite positive) matrices. Note that even if the observations $Y^{(1)}, \dots, Y^{(r)}$ differ because of the noise $E^{(1)}, \dots, E^{(r)}$, $B^*$ and the noise structure $S^*$ are shared across repetitions.

**Notation** We write $\|\cdot\|$ (*resp.* $\langle\cdot,\cdot\rangle$) for the Euclidean norm (*resp.* inner product) on vectors and matrices, $\|\cdot\|_p$ for the $\ell_p$ norm, for any $p \in [1, \infty)$. For a matrix $B \in \mathbb{R}^{p \times q}$, $\|B\|_{2,1} = \sum_{j=1}^p \|B_{j:}\|$ (*resp.* $\|B\|_{2,\infty} = \max_{j \in [p]} \|B_{j:}\|$), and for any $p \in [1, \infty]$, we write $\|B\|_{\mathscr{S},p}$ for the Schatten $p$-norm (*i.e.,* the $\ell_p$ norm of the singular values of B). The unit $\ell_p$ ball is written $\mathcal{B}_p$, $p \in [1, \infty)$. For $S_1$ and $S_2 \in \mathcal{S}_+^n$, $S_1 \succeq S_2$ if $S_1 - S_2 \in \mathcal{S}_+^n$. When we write $S_1 \succeq S_2$ we implicitly assume that both matrices belong to $\mathcal{S}_+^n$. For a square matrix $A \in \mathbb{R}^{n \times n}$, $\mathrm{Tr}(A)$ represents the trace of $A$ and $\|A\|_S = \sqrt{\mathrm{Tr}(A^\top SA)}$ is the Mahalanobis norm induced by $S \in \mathcal{S}_{++}^n$. For $a, b \in \mathbb{R}$, we denote $(a)_+ = \max(a, 0)$, $a \vee b = \max(a, b)$ and $a \wedge b = \min(a, b)$. The block soft-thresholding operator at level $\tau > 0$, is denoted $\mathrm{BST}(\cdot, \tau)$, and reads for any vector $x$, $\mathrm{BST}(x, \tau) = (1 - \tau/\|x\|)_+ x$. The identity matrix of size $n \times n$ is denoted $\mathrm{Id}_n$, and $[r]$ is the set of integers from 1 to $r$.

## 2.1 The proposed CLaR estimator

To leverage the multiple repetitions while taking into account the noise structure, we introduce the Concomitant Lasso with Repetitions (CLaR):

**Definition 1.** CLaR estimates the parameters of Model (1) by solving:

$$(\hat{B}^{\mathrm{CLaR}}, \hat{S}^{\mathrm{CLaR}}) \in \arg\min_{\substack{B \in \mathbb{R}^{p \times q} \\ S \succeq \underline{\sigma}\,\mathrm{Id}_n}} f(B, S) + \lambda \|B\|_{2,1}\,, \text{ with } f(B, S) \triangleq \sum_{l=1}^r \frac{\|Y^{(l)} - XB\|_{S^{-1}}^2}{2nqr} + \frac{\mathrm{Tr}(S)}{2n}\,,$$

(2)

where $\lambda > 0$ controls the sparsity of $\hat{B}^{\mathrm{CLaR}}$ and $\underline{\sigma} > 0$ controls the smallest eigenvalue of $\hat{S}^{\mathrm{CLaR}}$.

## 2.2 Connections with concomitant Lasso on averaged data

In low SNR settings, a standard way to deal with strong noise is to use the averaged observation $\bar{Y} \in \mathbb{R}^{n \times q}$ instead of the raw observations. The associated model reads:

$$\bar{Y} = XB^* + \tilde{S}^*\tilde{E}\,,$$

(3)

with $\tilde{S}^* \triangleq S^*/\sqrt{r}$ and $\tilde{E}$ has *i.i.d.* entries drawn from a standard normal distribution. The SNR[2] is multiplied by $\sqrt{r}$, yet the number of samples goes from $rnq$ to $nq$, making it statistically difficult to estimate the $\mathcal{O}(n^2)$ parameters of $S^*$. CLaR generalizes the Smoothed Generalized Concomitant Lasso (Massias et al., 2018a), which has the drawback of only targeting averaged observations:

**Definition 2** (SGCL, Massias et al. 2018a)**.** SGCL estimates the parameters of Model (3), by solving:

$$(\hat{B}^{\mathrm{SGCL}}, \hat{S}^{\mathrm{SGCL}}) \in \arg\min_{\substack{B \in \mathbb{R}^{p \times q} \\ \tilde{S} \succeq \underline{\sigma}/\sqrt{r}\,\mathrm{Id}_n}} \tilde{f}(B, \tilde{S}) + \lambda \|B\|_{2,1}\,, \text{ with } \tilde{f}(B, \tilde{S}) \triangleq \frac{\|\bar{Y} - XB\|_{\tilde{S}^{-1}}^2}{2nq} + \frac{\mathrm{Tr}(\tilde{S})}{2n}\,.$$

(4)

*Remark* 3. Note that $\hat{S}^{\mathrm{CLaR}}$ estimates $S^*$, while $\hat{S}^{\mathrm{SGCL}}$ estimates $\tilde{S}^* = S^*/\sqrt{r}$. Since we impose the constraint $\hat{S}^{\mathrm{CLaR}} \succeq \underline{\sigma}\,\mathrm{Id}_n$, we rescale the constraint so that $\hat{S}^{\mathrm{SGCL}} \succeq \underline{\sigma}/\sqrt{r}\,\mathrm{Id}_n$ in (4) for future comparisons. Also note that CLaR and SGCL are the same when $r = 1$ and $Y^{(1)} = \bar{Y}$.

The justification for CLaR is the following: if the quadratic loss $\|Y - XB\|^2$ were used, the parameters of Model (1) could be estimated by using either $\|\bar{Y} - XB\|^2$ or $\frac{1}{r}\sum\|Y^{(l)} - XB\|^2$ as a data-fitting term. Yet, both alternatives yield the same solutions as the two terms are equal up to constants. Hence, the quadratic loss does not leverage the multiple repetitions and ignores the noise structure. On the contrary, the more refined data-fitting term of CLaR allows to take into account the individual repetitions, leading to improved performance in applications.

# 3 Results and properties of CLaR

We start this part by introducing some elements of smoothing theory (Moreau, 1965; Nesterov, 2005; Beck and Teboulle, 2012) that sheds some light on the origin of the data-fitting term introduced earlier.

### 3.1 Smoothing of the nuclear norm

Let us analyze the data-fitting term of CLaR, by connecting it to the Schatten 1-norm. We derive a formula for the smoothing of the this norm (Proposition 4), which paves the way for a more general smoothing theory for matrix variables (see Appendix A). Let us define the following smoothing function:

$$\omega_{\underline{\sigma}}(\cdot) \triangleq \frac{1}{2}\left(\|\cdot\|^2 + n\right)\underline{\sigma} \ , \tag{5}$$

and the inf-convolution of functions $f_1$ and $f_2$, $f_1 \square f_2(y) \triangleq \inf_x f_1(x) + f_2(y-x)$. The name "smoothing" used in this paper comes from the following fact: if $f_1$ is a closed proper convex function, then $f_1^* + \frac{1}{2}\|\cdot\|^2$ is strongly convex, and thus its Fenchel transform $(f_1^* + \frac{1}{2}\|\cdot\|^2)^* = (f_1^* + (\frac{1}{2}\|\cdot\|^2)^*)^* = (f_1 \square \frac{1}{2}\|\cdot\|^2)^{**} = f_1 \square \frac{1}{2}\|\cdot\|^2$ is smooth (see Appendix A.1 for a detailed proof).

The next propositions are key to our framework and show the connection between the SGCL, CLaR and the Schatten 1-norm:

**Proposition 4** (Proof in Appendix A.3). The $\omega_{\underline{\sigma}}$-smoothing of the Schatten-1 norm, *i.e.,* the function $\|\cdot\|_{\mathscr{S},1}\square\omega_{\underline{\sigma}} : \mathbb{R}^{n\times q} \mapsto \mathbb{R}$, is the solution of the following smooth optimization problem:

$$(\|\cdot\|_{\mathscr{S},1}\square\omega_{\underline{\sigma}})(Z) = \min_{S \succeq \underline{\sigma}\operatorname{Id}_n} \tfrac{1}{2}\|Z\|^2_{S^{-1}} + \tfrac{1}{2}\operatorname{Tr}(S) \ . \tag{6}$$

Moreover $(\|\cdot\|_{\mathscr{S},1}\square\omega_{\underline{\sigma}})$ is a $\underline{\sigma}$-smooth $\frac{n}{2}\underline{\sigma}$-approximation of $\|\cdot\|_{\mathscr{S},1}$.

**Definition 5** (Clipped Square Root). For $\Sigma \in \mathcal{S}^n_+$ with spectral decomposition $\Sigma = U\operatorname{diag}(\gamma_1,\dots,\gamma_n)U^\top$ ($U$ is orthogonal), let us define the *Clipped Square Root* operator:

$$\operatorname{ClSqrt}(\Sigma,\underline{\sigma}) = U\operatorname{diag}(\sqrt{\gamma_1}\vee\underline{\sigma},\dots,\sqrt{\gamma_n}\vee\underline{\sigma})U^\top \ . \tag{7}$$

**Proposition 6** (Proof in Appendix B.1). Any solution of the CLaR Problem (2), $(\hat{\mathrm{B}},\hat{S}) = (\hat{\mathrm{B}}^{\mathrm{CLaR}},\hat{S}^{\mathrm{CLaR}})$ is also a solution of:

$$\hat{\mathrm{B}} = \underset{\mathrm{B}\in\mathbb{R}^{p\times q}}{\arg\min}\left(\|\cdot\|_{\mathscr{S},1}\square\omega_{\underline{\sigma}}\right)(Z) + \lambda n\|\mathrm{B}\|_{2,1}$$

$$\hat{S} = \operatorname{ClSqrt}\left(\tfrac{1}{rq}\hat{R}\hat{R}^\top,\underline{\sigma}\right) \ , \text{ where } \hat{R} = [Y^{(1)} - X\hat{\mathrm{B}}|\dots|Y^{(r)} - X\hat{\mathrm{B}}] \ .$$

Properties similar to Proposition 6 can be traced back to van de Geer and Stucky (2016, Sec 2.2), who introduced the multivariate square-root Lasso:

$$\hat{\mathrm{B}} \in \underset{\mathrm{B}\in\mathbb{R}^{p\times q}}{\arg\min}\frac{1}{n\sqrt{q}}\|\bar{Y} - X\mathrm{B}\|_{\mathscr{S},1} + \lambda\|\mathrm{B}\|_{2,1} \ , \tag{8}$$

and showed that if $(\bar{Y} - X\hat{\mathrm{B}})(\bar{Y} - X\hat{\mathrm{B}})^\top \succ 0$, the latter optimization problem admits a variational[3] formulation:

$$(\hat{\mathrm{B}},\hat{S}) \in \underset{\substack{\mathrm{B}\in\mathbb{R}^{p\times q}, \\ \tilde{S}\succ 0}}{\arg\min}\frac{1}{2nq}\|\bar{Y} - X\mathrm{B}\|^2_{S^{-1}} + \frac{\operatorname{Tr}(S)}{2n} + \lambda\|\mathrm{B}\|_{2,1} \ . \tag{9}$$

In other words Proposition 6 generalizes van de Geer (2016, Lemma 3.4) for all matrices $\bar{Y} - X\hat{\mathrm{B}}$, getting rid of the condition $(\bar{Y} - X\hat{\mathrm{B}})(\bar{Y} - X\hat{\mathrm{B}})^\top \succ 0$. In the present contribution, the problem formulation in Proposition 4 is motivated by computational aspects, as it helps to address the combined non-smoothness of the data-fitting term $\|\cdot\|_{\mathscr{S},1}$ and the penalty term $\|\cdot\|_{2,1}$. Note that another smoothing of the nuclear norm was proposed in Argyriou et al. (2008); Bach et al. (2012, Sec. 5.2):

$$Z \mapsto \min_{S\succ 0}\frac{1}{2}\operatorname{Tr}[Z^\top S^{-1}Z] + \frac{1}{2}\operatorname{Tr}(S) + \frac{\sigma^2}{2}\operatorname{Tr}(S^{-1}) \ , \tag{10}$$

which is a $\underline{\sigma}$-smooth $n\underline{\sigma}$-approximation of $\|\cdot\|_{\mathscr{S},1}$ (see Appendix A.5), therefore less precise than ours.

**Algorithm 1** ALTERNATE MINIMIZATION FOR CLAR
___
**input** : $X, \bar{Y}, \underline{\sigma}, \lambda, T_{S \text{ update}}, T$
**init** : $\mathrm{B} = 0_{p,q}, S^{-1} = \underline{\sigma}^{-1} \operatorname{Id}_n, \bar{R} = \bar{Y}, \operatorname{cov}_Y = \frac{1}{r} \sum_{l=1}^r Y^{(l)} Y^{(l)\top}$ // `precomputed`
**for** $t = 1, \dots, T$ **do**
&emsp; **if** $t = 1 \pmod{T_{S \text{ update}}}$ **then** // `noise update`
&emsp;&emsp; $RR^\top = \operatorname{RRT}(\operatorname{cov}_Y, Y, X, \mathrm{B})$ // `Eq. (15)`
&emsp;&emsp; $S \leftarrow \operatorname{ClSqrt}(\frac{1}{qr} RR^\top, \underline{\sigma})$ // `Eq. (12)`
&emsp;&emsp; **for** $j = 1, \dots, p$ **do** $L_j = X_{:j}^\top S^{-1} X_{:j}$
&emsp; **for** $j = 1, \dots, p$ **do** // `coef. update`
&emsp;&emsp; $\bar{R} \leftarrow \bar{R} + X_{:j} \mathrm{B}_{j:}$ ; $\mathrm{B}_{j:} \leftarrow \operatorname{BST}\left(\frac{X_{:j}^\top S^{-1} \bar{R}}{L_j}, \frac{\lambda n q}{L_j}\right)$ ; $\bar{R} \leftarrow \bar{R} - X_{:j} \mathrm{B}_{j:}$
**return** $\mathrm{B}, S$

---

Other alternatives to exploit the multiple repetitions without simply averaging them, would consist in investigating other Schatten $p$-norms:

$$\arg\min_{\mathrm{B} \in \mathbb{R}^{p \times q}} \frac{1}{\sqrt{rq}} \|[Y^{(1)} - X\mathrm{B}| \dots |Y^{(r)} - X\mathrm{B}]\|_{\mathscr{S},p} + \lambda n \|\mathrm{B}\|_{2,1} \quad . \tag{11}$$

Without smoothing, problems of the form given in Equation (11) present the drawback of having two non-smooth terms, and calling for primal-dual algorithms (Chambolle and Pock, 2011) with costly proximal operators. Even if the non-smooth Schatten 1-norm is replaced by the formula in Equation (6), numerical challenges remain: $S$ can approach 0 arbitrarily, hence, the gradient *w.r.t.* $S$ of the data-fitting term is not Lipschitz over the optimization domain. Recently, Molstad (2019) proposed two algorithms to directly solve Equation (11): a prox-linear ADMM, and accelerated proximal gradient descent, the latter lacking convergence guarantees since the composite objective has two non-smooth terms. Before that, van de Geer and Stucky (2016) devised a fixed point method, lacking descent guarantees. A similar problem was raised for the concomitant Lasso by Ndiaye et al. (2017) who used smoothing techniques to address it. Here we replaced the nuclear norm ($p = 1$) by its smoothed version $\|\cdot\|_{\mathscr{S},1} \square \omega_{\underline{\sigma}}$. Similar results for the Schatten 2-norm and Schatten $\infty$-norm are provided in the Appendix (Propositions 21 and 22).

### 3.2 Algorithmic details: convexity, (block) coordinate descent, parameters influence

We detail the principal results needed to solve Problem (2) numerically, leading to the implementation proposed in Algorithm 1. We first recall useful results for alternate minimization of convex composite problems.

**Proposition 7** (Proof in Appendix B.2). CLaR is jointly convex in $(\mathrm{B}, S)$. Moreover, $f$ is convex and smooth on the feasible set, and $\|\cdot\|_{2,1}$ is convex and separable in $\mathrm{B}_{j:}$'s, thus minimizing the objective alternatively in $S$ and in $\mathrm{B}_{j:}$'s (see Algorithm 1) converges to a global minimum.

Hence, for our alternate minimization implemenation, we only need to consider solving problems with B or $S$ fixed, which we detail in the next propositions.

**Proposition 8** (Minimization in $S$; proof in Appendix B.3). Let $\mathrm{B} \in \mathbb{R}^{n \times q}$ be fixed. The minimization of $f(\mathrm{B}, S)$ *w.r.t.* $S$ with the constraint $S \succeq \underline{\sigma} \operatorname{Id}_n$ admits the closed-form solution:

$$S = \operatorname{ClSqrt}\left(\frac{1}{rq} \sum_{l=1}^r (Y^{(l)} - X\mathrm{B})(Y^{(l)} - X\mathrm{B})^\top, \underline{\sigma}\right) \quad . \tag{12}$$

**Proposition 9** (Proof in Appendix B.4). For a fixed $S \in \mathcal{S}_{++}^n$, each step of the block minimization of $f(\cdot, S) + \lambda \|\cdot\|_{2,1}$ in the $j^{th}$ line of B admits a closed-form solution:

$$\mathrm{B}_{j:} = \operatorname{BST}\left(\mathrm{B}_{j:} + \frac{X_{:j}^\top S^{-1}(\bar{Y} - X\mathrm{B})}{\|X_{:j}\|_{S^{-1}}^2}, \frac{\lambda n q}{\|X_{:j}\|_{S^{-1}}^2}\right) \quad . \tag{13}$$

As for other Lasso-type estimators, there exists $\lambda_{\max} \geq 0$ such that whenever $\lambda \geq \lambda_{\max}$, the estimated coefficients vanish. This $\lambda_{\max}$ helps calibrating roughly $\lambda$ in practice by choosing it as a fraction of $\lambda_{\max}$.

**Proposition 10** (Critical regularization parameter; proof in Appendix B.5.)**.** For the CLaR estimator we have: with $S_{\max} \triangleq \mathrm{ClSqrt}\left(\frac{1}{qr}\sum_{l=1}^{r}Y^{(l)}Y^{(l)\top}, \underline{\sigma}\right)$,

$$\forall \lambda \geq \lambda_{\max} \triangleq \frac{1}{nq}\left\|X^\top S_{\max}^{-1}\bar{Y}\right\|_{2,\infty}, \quad \hat{\mathrm{B}}^{\mathrm{CLaR}} = 0 \ . \tag{14}$$

*Convex formulation benefits.* Thanks to the convex formulation, convergence of Algorithm 1 can be ensured using the duality gap as a stopping criterion (as it guarantees a targeted sub-optimality level). To compute the duality gap, we derive the dual of Problem (2) in Proposition 24. In addition, convexity allows to leverage acceleration methods such as working sets strategies (Fan and Lv, 2008; Tibshirani et al., 2012; Johnson and Guestrin, 2015; Massias et al., 2018b) or safe screening rules (El Ghaoui et al., 2012; Fercoq et al., 2015) while retaining theoretical convergence guarantees. Such techniques are trickier to adapt in the non-convex case (see Appendix C), as they could change the local minima reached.

*Choice of $\underline{\sigma}$.* Although $\underline{\sigma}$ has a smoothing interpretation, from a practical point of view it remains an hyperparameter to set. As in Massias et al. (2018a), $\underline{\sigma}$ is always chosen as follows: $\underline{\sigma} = \|Y\|/(1000 \times nq)$. In practice, the experimental results were little affected by the choice of $\underline{\sigma}$.

*Remark* 11. Once $\mathrm{cov}_Y \triangleq \frac{1}{r}\sum_1^r Y^{(l)}Y^{(l)\top}$ is pre-computed, the cost of updating $S$ does not depend on $r$, *i.e.,* is the same as working with averaged data. Indeed, with $R = [Y^{(1)} - X\mathrm{B}|\ldots|Y^{(r)} - X\mathrm{B}]$, the following computation can be done in $\mathcal{O}(qn^2)$ (details are in Appendix B.7).

$$RR^\top = \mathrm{RRT}(\mathrm{cov}_Y, Y, X, \mathrm{B}) \triangleq r\mathrm{cov}_Y + r(X\mathrm{B})(X\mathrm{B})^\top - r\bar{Y}^\top(X\mathrm{B}) - r(X\mathrm{B})^\top\bar{Y} \ . \tag{15}$$

Statistical properties showing the advantages of using CLaR (over SGCL) can be found in Appendix B.8. As one could expect, using $r$ times more observations improves the covariance estimation.

# 4 Experiments

Our Python code (with Numba compilation, Lam et al. 2015) is released as an open source package: https://github.com/QB3/CLaR. We compare CLaR to other estimators: SGCL (Massias et al., 2018a), an $\ell_{2,1}$ version of MLE (Chen and Banerjee, 2017; Lee and Liu, 2012) ($\ell_{2,1}$-MLE), a version of the $\ell_{2,1}$-MLE with multiple repetitions ($\ell_{2,1}$-MLER), an $\ell_{2,1}$ penalized version of MRCE (Rothman et al., 2010) with repetitions ($\ell_{2,1}$-MRCER) and the Multi-Task Lasso (MTL, Obozinski et al. 2010). The cost of an epoch of block coordinate descent is summarized in Table 1 in Appendix C.4 for each algorithm[4]. All competitors are detailed in Appendix C.

**Synthetic data** Here we demonstrate the ability of our estimator to recover the support *i.e.,* the ability to identify the predictive features. There are $n = 150$ observations, $p = 500$ features, $q = 100$ tasks. The design $X$ is random with Toeplitz-correlated features with parameter $\rho_X = 0.6$ (correlation between $X_{:i}$ and $X_{:j}$ is $\rho_X^{|i-j|}$), and its columns have unit Euclidean norm. The true coefficient $\mathrm{B}^*$ has 30 non-zeros rows whose entries are independent and normally centered distributed. $S^*$ is a Toeplitz matrix with parameter $\rho_S$. The SNR is fixed and constant across all repetitions

$$\mathrm{SNR} \triangleq \|X\mathrm{B}^*\|/\sqrt{r}\|X\mathrm{B}^* - \bar{Y}\| \ . \tag{16}$$

For Figures 1 to 3, the figure of merit is the ROC curve, *i.e.,* the true positive rate (TPR) against the false positive rate (FPR). For each estimator, the ROC curve is obtained by varying the value of the regularization parameter $\lambda$ on a geometric grid of 160 points, from $\lambda_{\max}$ (specific to each algorithm) to $\lambda_{\min}$, the latter also being estimator specific and chosen to obtain a FPR larger than $0.4$.

*Influence of noise structure.* Figure 1 represents the ROC curves for different values of $\rho_S$. As $\rho_S$ increases, the noise becomes more and more correlated. From left to right, the performance of CLaR, SGCL, $\ell_{2,1}$-MRCER, $\ell_{2,1}$-MRCE, and $\ell_{2,1}$-MLER increases as they are designed to exploit correlations in the noise, while the performance of MTL decreases, as its i.i.d. Gaussian noise model becomes less and less valid.

*Influence of SNR.* On Figure 2 we can see that when the SNR is high (left), all estimators (except $\ell_{2,1}$-MLE) reach the $(0, 1)$ point. This means that for each algorithm (except $\ell_{2,1}$-MLE), there exists a

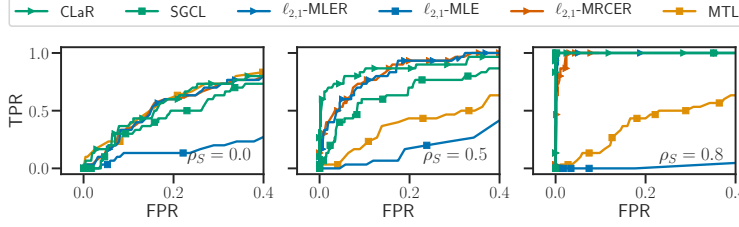

Figure 1 – *Influence of noise structure.* ROC curves of support recovery ($\rho_X = 0.6$, SNR $= 0.03$, $r = 20$) for different $\rho_S$ values.

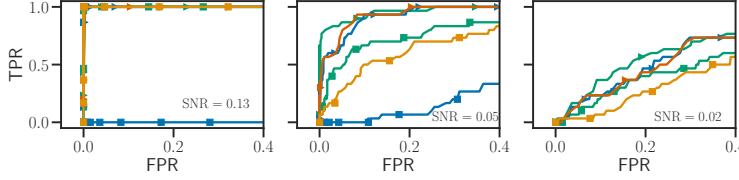

Figure 2 – *Influence of SNR.* ROC curves of support recovery ($\rho_X = 0.6$, $\rho_S = 0.4$, $r = 20$) for different SNR values.

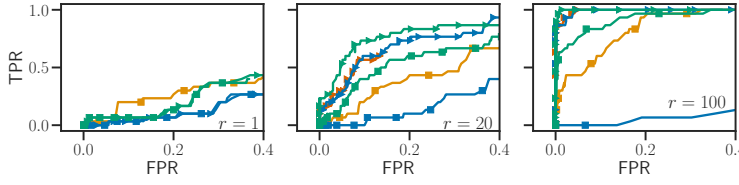

Figure 3 – *Influence of the number of repetitions.* ROC curves of support recovery ($\rho_X = 0.6$, SNR $= 0.03$, $\rho_S = 0.4$) for different $r$ values.

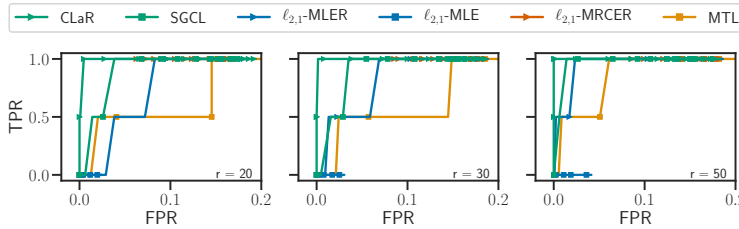

Figure 4 – *Influence of the number of repetitions.* ROC curves with empirical $X$ and $S$ and simulated B* (amp $= 2$ nA.m), for different number of repetitions.

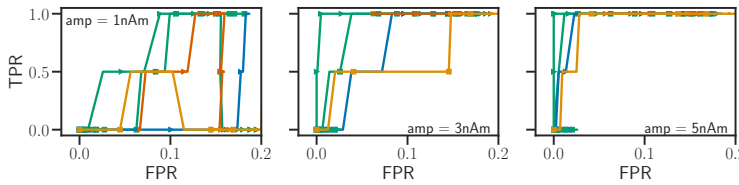

Figure 5 – *Amplitude influence.* ROC curves with empirical $X$ and $S$ and simulated B* ($r = 50$), for different amplitudes of the signal.

$\lambda$ such that the estimated support is exactly the true one. However, when the SNR decreases (middle), the performance of SGCL and MTL starts to drop, while that of CLaR, $\ell_{2,1}$-MLER and $\ell_{2,1}$-MRCER remains stable (CLaR performing better), highlighting their capacity to leverage multiple repetitions of measurements to handle the noise structure. Finally, when the SNR is too low (right), all algorithms perform poorly, but CLaR, $\ell_{2,1}$-MLER and $\ell_{2,1}$-MRCER still performs better.

*Influence of the number of repetitions.* Figure 3 shows ROC curves of all compared approaches for different $r$, starting from $r = 1$ (left) to 100 (right). Even with $r = 20$ (middle) CLaR outperforms the other estimators, and when $r = 100$ CLaR can better leverage the large number of repetitions.

**Realistic data** We now evaluate the estimators on realistic magneto- and electroencephalography (M/EEG) data. The M/EEG recordings measure the electrical potential and magnetic fields induced by the active neurons. Data are time series of length $q$ with $n$ sensors and $p$ sources mapping to locations in the brain. Because the propagation of the electromagnetic fields is driven by the linear Maxwell equations, one can assume that the relation between the measurements $Y^{(1)}, \ldots, Y^{(r)}$ and the amplitudes of sources in the brain B* is linear.

The M/EEG inverse problem consists in identifying B*. Because of the limited number of sensors (a few hundreds in practice), as well as the physics of the problem, the M/EEG inverse problem is severely ill-posed and needs to be regularized. Moreover, the experiments being usually short (less

than 1 s.) and focused on specific cognitive functions, the number of active sources is expected to be small, *i.e.,* $B^*$ is assumed to be row-sparse. This plausible biological assumption motivates the framework of Section 2 (Ou et al., 2009).

*Dataset.* We use the *sample* dataset [5] from the MNE software (Gramfort et al., 2014). The experimental conditions here are auditory stimulations in the right or left ear, leading to two main foci of activations in bilateral auditory cortices (*i.e.,* 2 non-zeros rows for $B^*$). For this experiment, we keep only the gradiometer magnetic channels. After removing one channel corrupted by artifacts, this leads to $n = 203$ signals. The length of the temporal series is $q = 100$, and the data contains $r = 50$ repetitions. We choose a source space of size $p = 1281$ which corresponds to about 1 cm distance between neighboring sources. The orientation is fixed, and normal to the cortical mantle.

*Realistic MEG data simulations.* We use here true empirical values for $X$ and $S$ by solving Maxwell equations and taking an empirical co-standard deviation matrix. To generate realistic MEG data we simulate neural responses $B^*$ with 2 non-zeros rows corresponding to areas known to be related to auditory processing (Brodmann area 22). Each non-zero row of $B^*$ is chosen as a sinusoidal signal with realistic frequency (5 Hz) and amplitude (amp $\sim 1 - 10$ nAm). We finally simulate $r$ MEG signals $Y^{(l)} = XB^* + S^*E^{(l)}$, $E^{(l)}$ being matrices with i.i.d. normal entries.

The signals being contaminated with correlated noise, if one wants to use homoscedastic solvers it is necessary to whiten the data first (and thus to have an estimation of the covariance matrix, the later often being unknown). In this experiment we demonstrate that without this whitening process, the homoscedastic solver MTL fails, as well as solvers which does not take in account the repetitions: SGCL and $\ell_{2,1}$-MLE. In this scenario CLaR, $\ell_{2,1}$-MLER and $\ell_{2,1}$-MRCER do succeed in recovering the sources, CLaR leading to the best results. As for the synthetic data, Figures 4 and 5 are obtained by varying the estimator-specific regularization parameter $\lambda$ from $\lambda_{\max}$ to $\lambda_{\min}$ on a geometric grid.

*Amplitude influence.* Figure 5 shows ROC curves for different values of the amplitude of the signal. When the amplitude is high (right), all the algorithms perform well, however when the amplitude decreases (middle) only CLaR leads to good results, almost hitting the $(0, 1)$ corner. When the amplitude gets lower (left) all algorithms perform worse, CLaR still yielding the best results.

*Influence of the number of repetitions.* Figure 4 shows ROC curves for different number of repetitions $r$. When the number of repetitions is high (right, $r = 50$), the algorithms taking into account all the repetitions (CLaR, $\ell_{2,1}$-MLER, $\ell_{2,1}$-MRCER) perform best, almost hitting the $(0, 1)$ corner, whereas the algorithms which do not take into account all the repetitions ($\ell_{2,1}$-MLE, MTL, SGCL) perform poorly. As soon as the number of repetitions decreases (middle and left) the performances of all the algorithms except CLaR start dropping severely. CLaR is once again the algorithm taking the most advantage of the number of repetitions.

**Real data**    As before, we use the *sample* dataset, keeping only the magnetometer magnetic channels ($n = 102$ signals). We choose a source space of size $p = 7498$ (about 5 mm between neighboring sources). The orientation is fixed, and normal to the cortical mantle. As for realistic data, $X$ is the empirical design matrix, but this time we use the empirical measurements $Y^{(1)}, \ldots, Y^{(r)}$. The experiment are left or right auditory stimulations, extensive results for right auditory stimulations (*resp.* visual stimulations) can be found in Appendix D.3 (*resp.* Appendix D.4 and D.5). As two sources are expected (one in each hemisphere, in bilateral auditory cortices), we vary $\lambda$ by dichotomy between $\lambda_{\max}$ (returning 0 sources) and a $\lambda_{\min}$ (returning more than 2 sources), until finding a $\lambda$ giving exactly 2 sources. Results are provided in Figures 6 and 7. Running times of each algorithm are of the same order of magnitude and can be found in Appendix D.2.

*Comments on Figure 6, left auditory stimulations.* Sources found by the algorithms are represented by red spheres. SGCL, $\ell_{2,1}$-MLE and $\ell_{2,1}$-MRCER completely fail, finding sources that are not in the auditory cortices at all (SGCL sources are deep, thus not in the auditory cortices, and cannot be seen). MTL and $\ell_{2,1}$-MLER do find sources in auditory cortices, but only in one hemisphere (left for MTL and right for $\ell_{2,1}$-MLER). CLaR is the only one that finds one source in each hemisphere in the auditory cortices as expected.

*Comments on Figure 7, right auditory stimulations.* In this experiment we only keep $r = 33$ repetitions (out of 65 available) and it can be seen that only CLaR finds correct sources, MTL finds sources only in one hemisphere and all the other algorithms do find sources that are not in the

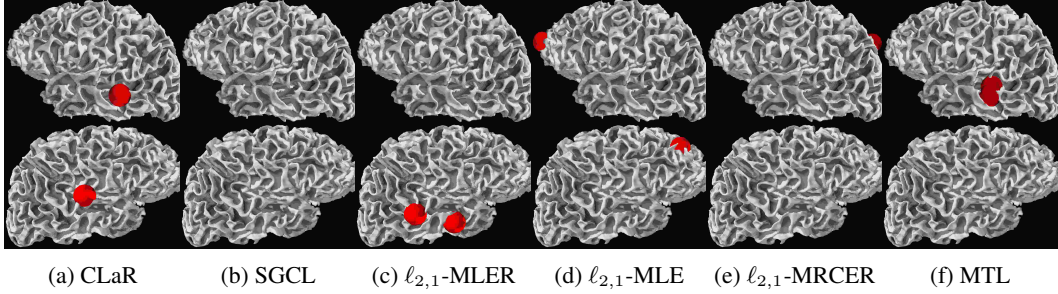

(a) CLaR     (b) SGCL     (c) $\ell_{2,1}$-MLER     (d) $\ell_{2,1}$-MLE     (e) $\ell_{2,1}$-MRCER     (f) MTL

Figure 6 – *Real data, left auditory stimulations ($n = 102$, $p = 7498$, $q = 76$, $r = 63$) Sources found in the left hemisphere (top) and the right hemisphere (bottom) after left auditory stimulations.*

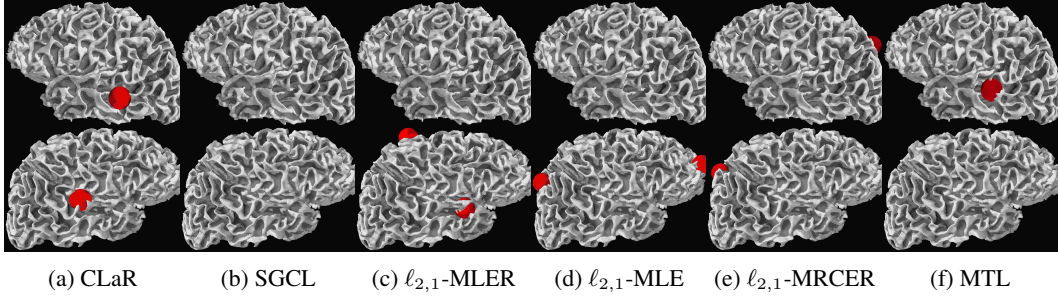

(a) CLaR     (b) SGCL     (c) $\ell_{2,1}$-MLER     (d) $\ell_{2,1}$-MLE     (e) $\ell_{2,1}$-MRCER     (f) MTL

Figure 7 – *Real data, right auditory stimulations ($n = 102$, $q = 7498$, $q = 76$, $r = 33$) Sources found in the left hemisphere (top) and the right hemisphere (bottom) after right auditory stimulations.*

auditory cortices. This highlights the robustness of CLaR, even with a limited number of repetitions, confirming previous experiments (see Figure 3).

**Conclusion** This work introduces CLaR, a sparse estimator for multitask regression. It is designed to handle correlated Gaussian noise in the context of repeated observations, a standard framework in applied sciences such as neuroimaging. The resulting optimization problem can be solved efficiently with state-of-the-art convex solvers, and the algorithmic cost is the same as for single repetition data. The theory of smoothing connects CLaR to the Schatten 1-Lasso in a principled manner, which opens the way to the use of more sophisticated datafitting terms. The benefits of CLaR for support recovery in the presence of non-white Gaussian noise were extensively evaluated against a large number of competitors, both on simulations and on empirical MEG data.

**Acknowledgments** This work was funded by ERC Starting Grant SLAB ERC-YStG-676943.

## Footnotes

[1]Other Schatten norms are treated in Appendix A.2.

[2]See the definition we consider in Eq. (16).

[3] also called *concomitant* formulation since minimization is performed over an additional variable (Owen, 2007; Ndiaye et al., 2017).

[4]The cost of computing the duality gap is also provided whenever available.

[5] publicly available real M/EEG data recorded after auditory or visual stimulations.

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
