[Supplementary Material · supplementary.pdf]

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

}$, $\|\mathrm{B}\|_{2,1} = \sum_{j=1}^p \|\mathrm{B}_{j:}\|$ (*resp.* $\|\mathrm{B}\|_{2,\infty} = \max_{j \in [p]} \|\mathrm{B}_{j:}\|$), and for any $p \in [1, \infty]$, we write $\|\mathrm{B}\|_{\mathscr{S},p}$ for the Schatten $p$-norm (*i.e.,* the $\ell_p$ norm of the singular values of B). The unit $\ell_p$ ball is written $\mathcal{B}_p$, $p \in [1, \infty)$. For $S_1$ and $S_2 \in \mathcal{S}_+^n$, $S_1 \succeq S_2$ if $S_1 - S_2 \in \mathcal{S}_+^n$. When we write $S_1 \succeq S_2$ we implicitly assume that both matrices belong to $\mathcal{S}_+^n$. For a square matrix $A \in \mathbb{R}^{n \times n}$, $\mathrm{Tr}(A)$ represents the trace of $A$ and $\|A\|_S = \sqrt{\mathrm{Tr}(A^\top S A)}$ is the Mahalanobis norm induced by $S \in \mathcal{S}_{++}^n$. For $a, b \in \mathbb{R}$, we denote $(a)_+ = \max(a, 0)$, $a \vee b = \max(a, b)$ and $a \wedge b = \min(a, b)$. The block soft-thresholding operator at level $\tau > 0$, is denoted $\mathrm{BST}(\cdot, \tau)$, and reads for any vector $x$, $\mathrm{BST}(x, \tau) = (1 - \tau/\|x\|)_+ x$. The identity matrix of size $n \times n$ is denoted $\mathrm{Id}_n$, and $[r]$ is the set of integers from 1 to $r$.

## 2.1   The proposed CLaR estimator

To leverage the multiple repetitions while taking into account the noise structure, we introduce the Concomitant Lasso with Repetitions (CLaR):

**Definition 1.** CLaR estimates the parameters of Model (1) by solving:

$$(\hat{\mathrm{B}}^{\mathrm{CLaR}}, \hat{\mathrm{S}}^{\mathrm{CLaR}}) \in \underset{\substack{\mathrm{B} \in \mathbb{R}^{p \times q} \\ S \succeq \underline{\sigma} \mathrm{Id}_n}}{\arg\min} f(\mathrm{B}, S) + \lambda \|\mathrm{B}\|_{2,1}, \text{ with } f(\mathrm{B}, S) \triangleq \sum_{l=1}^r \frac{\|Y^{(l)} - X\mathrm{B}\|_{S^{-1}}^2}{2nqr} + \frac{\mathrm{Tr}(S)}{2n},$$

(2)

where $\lambda > 0$ controls the sparsity of $\hat{\mathrm{B}}^{\mathrm{CLaR}}$ and $\underline{\sigma} > 0$ controls the smallest eigenvalue of $\hat{\mathrm{S}}^{\mathrm{CLaR}}$.

## 2.2   Connections with concomitant Lasso on averaged data

In low SNR settings, a standard way to deal with strong noise is to use the averaged observation $\bar{Y} \in \mathbb{R}^{n \times q}$ instead of the raw observations. The associated model reads:

$$\bar{Y} = X\mathrm{B}^* + \tilde{S}^* \tilde{\mathrm{E}} \ ,$$

(3)

with $\tilde{S}^* \triangleq S^* / \sqrt{r}$ and $\tilde{\mathrm{E}}$ has *i.i.d.* entries drawn from a standard normal distribution. The SNR[2] is multiplied by $\sqrt{r}$, yet the number of samples goes from $rnq$ to $nq$, making it statistically difficult to estimate the $\mathcal{O}(n^2)$ parameters of $S^*$. CLaR generalizes the Smoothed Generalized Concomitant Lasso (Massias et al., 2018a), which has the drawback of only targeting averaged observations:

**Definition 2** (SGCL, Massias et al. 2018a)**.** SGCL estimates the parameters of Model (3), by solving:

$$(\hat{\mathrm{B}}^{\mathrm{SGCL}}, \hat{\mathrm{S}}^{\mathrm{SGCL}}) \in \underset{\substack{\mathrm{B} \in \mathbb{R}^{p \times q} \\ \tilde{S} \succeq \underline{\sigma}/\sqrt{r} \mathrm{Id}_n}}{\arg\min} \tilde{f}(\mathrm{B}, \tilde{S}) + \lambda \|\mathrm{B}\|_{2,1}, \text{ with } \tilde{f}(\mathrm{B}, \tilde{S}) \triangleq \frac{\|\bar{Y} - X\mathrm{B}\|_{\

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

[6] As a reminder, for a scalar $t > 0$, the proximal operator of a function $h : \mathbb{R}^d \to \mathbb{R}$ can be defined for any $x_0 \in \mathbb{R}^d$ by $\operatorname{prox}_{t,h}(x_0) = \arg\min_{x \in \mathbb{R}^d} \frac{1}{2t}\|x - x_0\|^2 + h(x) .$

[7]In that case we plug $\hat{B} = \hat{B}^{\mathrm{CLaR}}$ (resp. $\hat{B} = \hat{B}^{\mathrm{CLaR}}$) in Proposition 25.

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

# A  Smoothing theory for convex optimization

**Notation**  Let $d \in \mathbb{N}$, and let $\mathcal{C}$ be a closed and convex subset of $\mathbb{R}^d$. We write $\iota_{\mathcal{C}}$ for the indicator function of the set $\mathcal{C}$, *i.e.*, $\iota_{\mathcal{C}}(x) = 0$ if $x \in \mathcal{C}$ and $\iota_{\mathcal{C}}(x) = +\infty$ otherwise, and $\Pi_{\mathcal{C}}$ for the Euclidean projection on $\mathcal{C}$. The Fenchel conjugate of a function $h : \mathbb{R}^d \to \mathbb{R}$ is written $h^*$ and is defined for any $y \in \mathbb{R}^d$, by $h^*(y) = \sup_{x \in \mathbb{R}^d} \langle x, y \rangle - h(x)$. For $p \in [1, +\infty]$, let us write $\mathcal{B}_{\mathscr{S},p}$ for the Schatten-$p$ unit ball, and $\|\cdot\|_p$ for the standard $\ell_p$-norm in $\mathbb{R}^d$.

## A.1  Basic properties of inf-convolution

**Proposition 12.** Let $g : \mathbb{R}^d \to \mathbb{R}$, $h : \mathbb{R}^d \to \mathbb{R}$ be closed proper convex functions. Then, the following holds (see Parikh et al. 2013, p. 136):

$$h^{**} = h \ , \tag{17}$$

$$(h \,\square\, g)^* = h^* + g^* \ , \tag{18}$$

$$\left( \underline{\sigma} g \left( \tfrac{\cdot}{\underline{\sigma}} \right) \right)^* = \underline{\sigma} g^* \ , \tag{19}$$

$$\|\cdot\|_p^* = \iota_{\mathcal{B}_{p^*}}, \ \text{where} \ \frac{1}{p} + \frac{1}{p^*} = 1 \ , \tag{20}$$

$$(h + \delta)^* = h^* - \delta, \quad \forall \delta \in \mathbb{R} \ , \tag{21}$$

$$\left( \tfrac{1}{2} \|\cdot\|^2 \right)^* = \tfrac{1}{2} \|\cdot\|^2 \ . \tag{22}$$

## A.2  Smoothing of Schatten norms

In all this section, the variable is a matrix $Z \in \mathbb{R}^{n \times q}$, and the function $\omega_{\underline{\sigma}}$ is defined as $\underline{\sigma} \omega \left( \tfrac{\cdot}{\underline{\sigma}} \right)$.

**Lemma 13.** Let $c \in \mathbb{R}, \in [1, \infty]$. Let $p^* \in [1, \infty]$ be the Hölder conjugate of $p$, $\frac{1}{p} + \frac{1}{p^*} = 1$. For the choice $\omega(\cdot) = \frac{1}{2} \|\cdot\|^2 + c$, the following holds true:

$$\left( \|\cdot\|_{\mathscr{S},p} \,\square\, \omega_{\underline{\sigma}} \right)(Z) = \frac{1}{2\underline{\sigma}} \|Z\|^2 + c\underline{\sigma} - \frac{\sigma}{2} \left\| \Pi_{\mathcal{B}_{\mathscr{S},p^*}} \left( \tfrac{Z}{\underline{\sigma}} \right) - \tfrac{Z}{\underline{\sigma}} \right\|^2 \ .$$

*Proof.*

$$
\begin{aligned}
\left( \|\cdot\|_{\mathscr{S},p} \,\square\, \omega_{\underline{\sigma}} \right)(Z) &= \left( \|\cdot\|_{\mathscr{S},p} \,\square\, \omega_{\underline{\sigma}} \right)^{**}(Z) && \text{(using Eq. (17))} \\
&= \left( \|\cdot\|_{\mathscr{S},p}^* + \omega_{\underline{\sigma}}^* \right)^*(Z) && \text{(using Eq. (18))} \\
&= \left( \iota_{\mathcal{B}_{\mathscr{S},p^*}} + \tfrac{\sigma}{2} \|\cdot\|^2 - c\underline{\sigma} \right)^*(Z) && \text{(using Eq. (20))} \\
&= \left( \tfrac{\sigma}{2} \|\cdot\|^2 + \iota_{\mathcal{B}_{\mathscr{S},p^*}} \right)^*(Z) + c\underline{\sigma} && \text{(using Eq. (21)) .} \tag{23}
\end{aligned}
$$

We can now compute the last Fenchel transform remaining:

$$\left(\frac{\sigma}{2}\|\cdot\|^2 + \iota_{\mathcal{B}_{\mathscr{S},p^*}}\right)^*(Z) = \sup_{U\in\mathbb{R}^{n\times q}}\left(\langle U,Z\rangle - \frac{\sigma}{2}\|U\|^2 - \iota_{\mathcal{B}_{\mathscr{S},p^*}}(U)\right)$$

$$= \sup_{U\in\mathcal{B}_{\mathscr{S},p^*}}\left(\langle U,Z\rangle - \frac{\sigma}{2}\|U\|^2\right)$$

$$= -\inf_{U\in\mathcal{B}_{\mathscr{S},p^*}}\left(\frac{\sigma}{2}\|U\|^2 - \langle U,Z\rangle\right)$$

$$= -\underline{\sigma}\cdot\inf_{U\in\mathcal{B}_{\mathscr{S},p^*}}\left(\frac{1}{2}\|U\|^2 - \left\langle U,\frac{Z}{\underline{\sigma}}\right\rangle\right)$$

$$= -\underline{\sigma}\cdot\inf_{U\in\mathcal{B}_{\mathscr{S},p^*}}\left(\frac{1}{2}\left\|U-\frac{Z}{\underline{\sigma}}\right\|^2 - \frac{1}{2\underline{\sigma}^2}\|Z\|^2\right)$$

$$= \frac{1}{2\underline{\sigma}}\|Z\|^2 - \frac{\sigma}{2}\cdot\inf_{U\in\mathcal{B}_{\mathscr{S},p^*}}\left(\left\|U-\frac{Z}{\underline{\sigma}}\right\|^2\right)$$

$$= \frac{1}{2\underline{\sigma}}\|Z\|^2 - \frac{\sigma}{2}\left\|\Pi_{\mathcal{B}_{\mathscr{S},p^*}}\left(\frac{Z}{\underline{\sigma}}\right) - \frac{Z}{\underline{\sigma}}\right\|^2 . \qquad (24)$$

The result follows by combining Eqs. (23) and (24).

$\square$

### A.3 Schatten 1-norm (nuclear/trace norm), proof of Proposition 4

#### A.3.1 Preliminary lemmas

First we need the formula of the projection of a matrix onto the Schatten infinity ball:

**Lemma 14** (Projection onto $\mathcal{B}_{\mathscr{S},\infty}$, Beck 2017, Example 7.31). Let $Z \in \mathbb{R}^{n\times q}$, let $Z = V\operatorname{diag}(\gamma_1,\dots,\gamma_{n\wedge q})W^\top$ be a singular value decomposition of $Z$, then:

$$\Pi_{\mathcal{B}_{\mathscr{S},\infty}}(Z) = V\operatorname{diag}(\gamma_1 \wedge 1,\dots,\gamma_{n\wedge q}\wedge 1)W^\top . \qquad (25)$$

Then we need to link the value of the primal to the singular values of $ZZ^\top$:

**Lemma 15** (Value of the primal). Let $\gamma_1,\dots,\gamma_{n\wedge q}$ be a singular value decomposition of $Z$, then:

i) $\min_{S\succeq\underline{\sigma}\operatorname{Id}_n}\frac{1}{2}\operatorname{Tr}[Z^\top S^{-1}Z]+\frac{1}{2}\operatorname{Tr}(S) = \frac{1}{2}\sum_{i=1}^{n\wedge q}\frac{\gamma_i^2}{\gamma_i\vee\underline{\sigma}}+\frac{1}{2}\sum_{i=1}^{n\wedge q}\gamma_i\vee\underline{\sigma}+\frac{1}{2}(n-n\wedge q)\underline{\sigma}$ ,

ii) $\min_{S\succeq\underline{\sigma}\operatorname{Id}_q}\frac{1}{2}\operatorname{Tr}[ZS^{-1}Z^\top]+\frac{1}{2}\operatorname{Tr}(S) = \frac{1}{2}\sum_{i=1}^{n\wedge q}\frac{\gamma_i^2}{\gamma_i\vee\underline{\sigma}}+\frac{1}{2}\sum_{i=1}^{n\wedge q}\gamma_i\vee\underline{\sigma}+\frac{1}{2}(q-n\wedge q)\underline{\sigma}$ .

*Proof of Lemma 15 i).* The minimum in the left hand side is attained at $\hat{S} = U\operatorname{diag}(\gamma_1\vee\underline{\sigma},\dots,\gamma_{n\wedge q}\vee\underline{\sigma},\underline{\sigma},\dots,\underline{\sigma})U^\top$ (see Massias et al. 2018a, Prop. 2).

$$\min_{S\succeq\sigma\operatorname{Id}_n}\frac{1}{2}\operatorname{Tr}[Z^\top S^{-1}Z]+\frac{1}{2}\operatorname{Tr}(S) = \frac{1}{2}\operatorname{Tr}[Z^\top \hat{S}^{-1}Z]+\frac{1}{2}\operatorname{Tr}(\hat{S})$$

$$= \frac{1}{2}\operatorname{Tr}[\hat{S}^{-1}ZZ^\top]+\frac{1}{2}\operatorname{Tr}(\hat{S})$$

$$= \frac{1}{2}\operatorname{Tr}[U\operatorname{diag}(\gamma_1^2/(\gamma_1\vee\underline{\sigma}),\dots,\gamma_{n\wedge q}^2/(\gamma_{n\wedge q}\vee\underline{\sigma}),0,\dots,0)U^\top]$$

$$+ \frac{1}{2}\operatorname{Tr}[U\operatorname{diag}(\gamma_1\vee\underline{\sigma},\dots,\gamma_{n\wedge q}\vee\underline{\sigma},\underline{\sigma},\dots,\underline{\sigma})U^\top]$$

$$= \frac{1}{2}\sum_{i=1}^{n\wedge q}\frac{\gamma_i^2}{\gamma_i\vee\underline{\sigma}}+\frac{1}{2}\sum_{i=1}^{n\wedge q}\gamma_i\vee\underline{\sigma}+\frac{1}{2}\sum_{n\wedge q+1}^{n}\underline{\sigma}$$

$$= \frac{1}{2}\sum_{i=1}^{n\wedge q}\frac{\gamma_i^2}{\gamma_i\vee\underline{\sigma}}+\frac{1}{2}\sum_{i=1}^{n\wedge q}\gamma_i\vee\underline{\sigma}+\frac{1}{2}(n-n\wedge q)\underline{\sigma} . \qquad (26)$$

This completes the proof of Lemma 15 *i)*. Lemma 15 *ii)* is obtained by symmetry. $\square$

### A.3.2 Main result: an explicit variational formula for the inf-convolution smoothing of the nuclear norm

We now recall the main result that we claim to prove:

**Proposition 4** (Proof in Appendix A.3). The $\omega_{\underline{\sigma}}$-smoothing of the Schatten-1 norm, *i.e.,* the function $\|\cdot\|_{\mathscr{S},1} \,\square\, \omega_{\underline{\sigma}} : \mathbb{R}^{n \times q} \mapsto \mathbb{R}$, is the solution of the following smooth optimization problem:

$$(\|\cdot\|_{\mathscr{S},1} \,\square\, \omega_{\underline{\sigma}})(Z) = \min_{S \succeq \underline{\sigma} \operatorname{Id}_n} \tfrac{1}{2} \|Z\|_{S^{-1}}^2 + \tfrac{1}{2} \operatorname{Tr}(S) \;. \tag{6}$$

Moreover $(\|\cdot\|_{\mathscr{S},1} \,\square\, \omega_{\underline{\sigma}})$ is a $\underline{\sigma}$-smooth $\frac{n}{2}\underline{\sigma}$-approximation of $\|\cdot\|_{\mathscr{S},1}$.

*Proof.* Let $V \operatorname{diag}(\gamma_1, \dots, \gamma_{n \wedge q}) W^\top$ be a singular value decomposition of $Z$. We remind that $\Pi_{\mathcal{B}_{\mathscr{S},\infty}}$, the projection over $\mathcal{B}_{\mathscr{S},\infty}$, is given by (see Beck 2017, Example 7.31):

$$\Pi_{\mathcal{B}_{\mathscr{S},\infty}} \left(\tfrac{Z}{\underline{\sigma}}\right) = V \operatorname{diag}\left(\Pi_{\mathcal{B}_{\mathscr{S},\infty}}\left(\tfrac{\gamma_1}{\underline{\sigma}}, \dots, \tfrac{\gamma_{n \wedge q}}{\underline{\sigma}}\right)\right) W^\top$$

$$= V \operatorname{diag}\left(\tfrac{\gamma_1}{\underline{\sigma}} \wedge 1, \dots, \tfrac{\gamma_{n \wedge q}}{\underline{\sigma}} \wedge 1\right) W^\top \;, \tag{27}$$

where we used that the (vectorial) projection over $\mathcal{B}_\infty$ is given coordinate-wise by $(\Pi_{\mathcal{B}_\infty}(\gamma_i))_i = (\gamma_i \wedge 1)_i$ on the positive orthant. Then we have,

$$\left\| \Pi_{\mathcal{B}_{\mathscr{S},\infty}}\left(\tfrac{Z}{\underline{\sigma}}\right) - \tfrac{Z}{\underline{\sigma}} \right\|^2 = \left\| V \operatorname{diag}\left(\tfrac{\gamma_1}{\underline{\sigma}} \wedge 1 - \tfrac{\gamma_1}{\underline{\sigma}}, \dots, \tfrac{\gamma_{n \wedge q}}{\underline{\sigma}} \wedge 1 - \tfrac{\gamma_{n \wedge q}}{\underline{\sigma}}\right) W^\top \right\|^2$$

$$= \sum_{i=1}^{n \wedge q} \left(\tfrac{\gamma_i}{\underline{\sigma}} \wedge 1 - \tfrac{\gamma_i}{\underline{\sigma}}\right)^2$$

$$= \frac{1}{\underline{\sigma}^2} \sum_{i=1}^{n \wedge q} (\gamma_i \wedge \underline{\sigma} - \gamma_i)^2$$

$$= \frac{1}{\underline{\sigma}^2} \sum_{\gamma_i > \underline{\sigma}} (\gamma_i \wedge \underline{\sigma} - \gamma_i)^2$$

$$= \frac{1}{\underline{\sigma}^2} \sum_{\gamma_i > \underline{\sigma}} (\underline{\sigma} - \gamma_i)^2$$

$$= \frac{1}{\underline{\sigma}^2} \sum_{\gamma_i > \underline{\sigma}} (\underline{\sigma}^2 + \gamma_i^2 - 2\underline{\sigma}\gamma_i)$$

$$= \sum_{\gamma_i > \underline{\sigma}} 1 + \frac{1}{\underline{\sigma}^2} \sum_{\gamma_i > \underline{\sigma}} \gamma_i^2 - 2\frac{1}{\underline{\sigma}} \sum_{\gamma_i > \underline{\sigma}} \gamma_i \tag{28}$$

$$-\frac{\sigma}{2} \left\| \Pi_{\mathcal{B}_{\mathscr{S},\infty}}\left(\tfrac{Z}{\underline{\sigma}}\right) - \tfrac{Z}{\underline{\sigma}} \right\|^2 = -\frac{\sigma}{2} \sum_{\gamma_i > \underline{\sigma}} 1 - \frac{1}{2\underline{\sigma}} \sum_{\gamma_i > \underline{\sigma}} \gamma_i^2 + \sum_{\gamma_i > \underline{\sigma}} \gamma_i \;. \tag{29}$$

By combining Lemma 13 and Eq. (29) with $p^* = \infty, c \in \mathbb{R}$, it follows:

$$\left(\|\cdot\|_{\mathscr{S},1} \,\square\, \omega_{\underline{\sigma}}\right)(Z) = \frac{1}{2\underline{\sigma}} \sum_{i=1}^{n} \gamma_i^2 + c\underline{\sigma} - \frac{\sigma}{2} \sum_{\gamma_i > \underline{\sigma}} 1 - \frac{1}{2\underline{\sigma}} \sum_{\gamma_i > \underline{\sigma}} \gamma_i^2 + \sum_{\gamma_i > \underline{\sigma}} \gamma_i$$

$$= \frac{1}{2\underline{\sigma}} \sum_{\gamma_i \leq \underline{\sigma}}^{n} \gamma_i^2 + c\underline{\sigma} - \frac{\sigma}{2} \sum_{\gamma_i > \underline{\sigma}} 1 + \sum_{\gamma_i > \underline{\sigma}} \gamma_i \qquad \text{by grouping the } \gamma_i \text{ terms}$$

$$= \frac{1}{2\underline{\sigma}} \sum_{\gamma_i \leq \underline{\sigma}}^{n} \gamma_i^2 + \sum_{\gamma_i > \underline{\sigma}} \gamma_i - \frac{\sigma}{2} \sum_{\gamma_i > \underline{\sigma}} 1 + c\underline{\sigma} \qquad \text{by reordering.}$$

$$\tag{30}$$

The goal is now to link the optimization problem to the right-hand side of Equation (30). Let $ZZ^\top = U^\top \operatorname{diag}(\underbrace{\gamma_1, \ldots, \gamma_{n \wedge q}, 0, \ldots, 0}_{\in \mathbb{R}^n})U$ be an eigenvalue decomposition of $ZZ^\top$.

$$\min_{S \succeq \underline{\sigma} \operatorname{Id}_n} \frac{1}{2} \operatorname{Tr}[Z^\top S^{-1} Z] + \frac{1}{2} \operatorname{Tr}(S) = \frac{1}{2} \sum_{i=1}^{n \wedge q} \frac{\gamma_i^2}{\gamma_i \vee \underline{\sigma}} + \frac{1}{2} \sum_{i=1}^{n \wedge q} \gamma_i \vee \underline{\sigma} + \frac{1}{2}(n - n \wedge q)\underline{\sigma} \quad \text{(using Lemma 15)}$$

$$= \frac{1}{2\underline{\sigma}} \sum_{\gamma_i \leq \underline{\sigma}} \gamma_i^2 + \frac{1}{2} \sum_{\gamma_i > \underline{\sigma}} \gamma_i + \frac{1}{2} \sum_{\gamma_i \leq \underline{\sigma}} \underline{\sigma} + \frac{1}{2} \sum_{\gamma_i > \underline{\sigma}} \gamma_i + \frac{1}{2}(n - n \wedge q)\underline{\sigma}$$

$$= \frac{1}{2\underline{\sigma}} \sum_{\gamma_i \leq \underline{\sigma}} \gamma_i^2 + \sum_{\gamma_i > \underline{\sigma}} \gamma_i + \frac{\underline{\sigma}}{2} \sum_{\gamma_i \leq \underline{\sigma}} 1 + \frac{1}{2}(n - n \wedge q)\underline{\sigma}$$

$$= \frac{1}{2\underline{\sigma}} \sum_{\gamma_i \leq \underline{\sigma}} \gamma_i^2 + \sum_{\gamma_i > \underline{\sigma}} \gamma_i + \frac{\underline{\sigma}}{2}\left(n \wedge q - \sum_{\gamma_i > \underline{\sigma}} 1\right) + \frac{1}{2}(n - n \wedge q)\underline{\sigma}$$

$$= \frac{1}{2\underline{\sigma}} \sum_{\gamma_i \leq \underline{\sigma}} \gamma_i^2 + \sum_{\gamma_i > \underline{\sigma}} \gamma_i - \frac{\underline{\sigma}}{2} \sum_{\gamma_i > \underline{\sigma}} 1 + \underbrace{\frac{\underline{\sigma}}{2} n \wedge q + \frac{1}{2}(n - n \wedge q)\underline{\sigma}}_{\frac{\underline{\sigma}}{2}n}$$

$$= \frac{1}{2\underline{\sigma}} \sum_{\gamma_i \leq \underline{\sigma}} \gamma_i^2 + \sum_{\gamma_i > \underline{\sigma}} \gamma_i - \frac{\underline{\sigma}}{2} \sum_{\gamma_i > \underline{\sigma}} 1 + \frac{\underline{\sigma}}{2} n \ , \tag{31}$$

and identifying Equations (30) and (31) leads to the result for $c = \frac{n}{2}$. □

### A.4 Properties of the proposed smoothing for the nuclear norm

First let us recall the definition of a smoothable function and a $\mu$-smooth approximation of Beck and Teboulle (2012, Def. 2.1):

**Definition 16** (Smoothable function, $\mu$-smooth approximation). Let $g : \mathbb{E} \to \,]-\infty, +\infty]$ be a closed and proper convex function, and let $E \subseteq \operatorname{dom}(g)$ be a closed convex set. The function $g$ is called $(\alpha, \delta, K)$-*smoothable* on $E$ if there exists $\delta_1, \delta_2$ satisfying $\delta_1 + \delta_2 = \delta > 0$ such that for every $\mu$ there exists a continuously differentiable convex function $g_\mu : \mathbb{E} \to \,]-\infty, +\infty[$ such that the following holds:

  i) $g(x) - \delta_1 \mu \leq g_\mu(x) \leq g(x) + \delta_2 \mu$ for every $x \in E$ .

  ii) The function $\nabla g_\mu$ has a Lipschitz constant which is less than or equal to $K + \frac{\alpha}{\mu}$:

$$\|\nabla g_\mu(x) - \nabla g_\mu(y)\| \leq \left(K + \frac{\alpha}{\mu}\right) \|x - y\| \text{ for every } x, y \in E \ . \tag{32}$$

The function $g$ is called a $\mu$-*smooth* approximation of $g$ with parameters $(\alpha, \delta, K)$.

The nuclear norm $\|\cdot\|_{\mathscr{S},1}$ is non-smooth (and not even differentiable at 0), but one can construct a smooth approximation of the nuclear norm based on the following variational formula, if $ZZ^\top \succ 0$:

$$\|Z\|_{\mathscr{S},1} = \min_{S \succ 0} \frac{1}{2} \operatorname{Tr}[Z^\top S^{-1} Z] + \frac{1}{2} \operatorname{Tr}(S) \ , \tag{33}$$

see van de Geer (2016, Lemma 3.4). When $ZZ^\top \not\succ 0$, one can approximate $\|\cdot\|_{\mathscr{S},1}$ with

$$\min_{S \succeq \underline{\sigma} \operatorname{Id}} \frac{1}{2} \operatorname{Tr}[Z^\top S^{-1} Z] + \frac{1}{2} \operatorname{Tr}(S) = \|\cdot\|_{\mathscr{S},1} \square \omega_{\underline{\sigma}} \ , \tag{34}$$

as shown in Appendix A.3. It can be shown that this approximation stays close to the nuclear norm.

**Proposition 17.** $\|\cdot\|_{\mathscr{S},1} \square \omega_{\underline{\sigma}}$ is a $\underline{\sigma}$-smooth approximation of $\|\cdot\|_{\mathscr{S},1}$ with parameters $(1, \frac{n}{2}, 0)$. More precisely: $\|\cdot\|_{\mathscr{S},1} \square \omega_{\underline{\sigma}}$ has a $\underline{\sigma}$-Lipschitz gradient and

$$0 \leq \|\cdot\|_{\mathscr{S},1} \square \omega_{\underline{\sigma}} - \|\cdot\|_{\mathscr{S},1} = \frac{\underline{\sigma}}{2} \sum_{\gamma_i < \underline{\sigma}} \left(1 - \frac{\gamma_i}{\underline{\sigma}}\right)^2 \leq \frac{\underline{\sigma}}{2} n \ . \tag{35}$$

*Proof.* Since $\omega$ is 1-smooth, Beck and Teboulle (2012, Thm. 4.1) shows that $\|\cdot\|_{\mathscr{S},1} \square \omega_{\underline{\sigma}}$ is $\underline{\sigma}$-smooth.

Let $Z \in \mathbb{R}^{n \times q}$ and let $\gamma_1, \ldots, \gamma_{n \wedge q}$ be its singular value decomposition:

$$
\begin{aligned}
\left(\|\cdot\|_{\mathscr{S},1} \square \omega_{\underline{\sigma}}\right)(Z) - \|Z\|_{\mathscr{S},1} &= \frac{1}{2}\sum_{i=1}^{n \wedge q} \frac{\gamma_i^2}{\gamma_i \vee \underline{\sigma}} + \frac{1}{2}\sum_{i=1}^{n \wedge q} \gamma_i \vee \underline{\sigma} + \frac{1}{2}\sum_{n \wedge q+1}^{n} \underline{\sigma} - \sum_{i=1}^{n \wedge q} \gamma_i \\
&= \frac{1}{2}\sum_{i=1}^{n \wedge q}\left(\frac{\gamma_i^2}{\gamma_i \vee \underline{\sigma}} + \gamma_i \vee \underline{\sigma} - 2\gamma_i\right) + \frac{1}{2}\sum_{n \wedge q+1}^{n} \underline{\sigma} \\
&= \frac{1}{2}\sum_{\gamma_i \le \underline{\sigma}}\left(\frac{\gamma_i^2}{\gamma_i \vee \underline{\sigma}} + \gamma_i \vee \underline{\sigma} - 2\gamma_i\right) + \frac{1}{2}\sum_{n \wedge q+1}^{n} \underline{\sigma} \\
&= \frac{1}{2}\sum_{\gamma_i \le \underline{\sigma}}\left(\frac{\gamma_i^2}{\underline{\sigma}} + \underline{\sigma} - 2\gamma_i\right) + \frac{1}{2}\sum_{n \wedge q+1}^{n} \underline{\sigma} \\
&= \frac{1}{2}\sum_{\gamma_i \le \underline{\sigma}} \frac{(\gamma_i - \underline{\sigma})^2}{\underline{\sigma}} + \frac{1}{2}(n - n \wedge q)\underline{\sigma} \ . \quad (36)
\end{aligned}
$$

Hence,

$$
0 \le \left(\|\cdot\|_{\mathscr{S},1} \square \omega_{\underline{\sigma}}\right)(Z) - \|Z\|_{\mathscr{S},1} = \frac{1}{2}\sum_{\gamma_i \le \underline{\sigma}} \frac{(\gamma_i - \underline{\sigma})^2}{\underline{\sigma}} + \frac{1}{2}(n - n \wedge q)\underline{\sigma} \le \frac{\underline{\sigma}}{2}n \ . \quad (37)
$$

Moreover this bound is attained when $Z = 0$. $\qquad \square$

### A.5 Comparison with another smoothing of the nuclear norm

Another regularization was proposed in Argyriou et al. (2008); Bach et al. (2012, p. 62):

$$
\min_{S \succ 0} \underbrace{\frac{1}{2}\operatorname{Tr}[Z^\top S^{-1} Z] + \frac{1}{2}\operatorname{Tr}(S) + \frac{\underline{\sigma}^2}{2}\operatorname{Tr}(S^{-1})}_{h(S^{-1})} \ . \quad (38)
$$

By putting the gradient of the objective function in Equation (38) to zero it follows that:

$$
0 = \nabla h(\hat{S}^{-1}) = ZZ^\top - \hat{S}^2 + \underline{\sigma}^2 \operatorname{Id} \ , \quad (39)
$$

leading to :

$$
\hat{S} = (ZZ^\top + \underline{\sigma}^2 \operatorname{Id})^{\frac{1}{2}} \ . \quad (40)
$$

Let $\gamma_1, \ldots, \gamma_{n \wedge q}$ be the singular values of $Z$:

$$
\begin{aligned}
\frac{1}{2}\operatorname{Tr}[Z^\top \hat{S}^{-1} Z] + \frac{1}{2}\operatorname{Tr}(\hat{S}) + \frac{\underline{\sigma}^2}{2}\operatorname{Tr}(\hat{S}^{-1}) &= \frac{1}{2}\sum_{i=1}^{n}\left(\frac{\gamma_i^2}{\sqrt{\gamma_i^2 + \underline{\sigma}^2}} + \sqrt{\gamma_i^2 + \underline{\sigma}^2} + \frac{\underline{\sigma}^2}{\sqrt{\gamma_i^2 + \underline{\sigma}^2}}\right) \\
&= \frac{1}{2}\sum_{i=1}^{n}\left(\frac{\gamma_i^2 + \gamma_i^2 + \underline{\sigma}^2 + \underline{\sigma}^2}{\sqrt{\gamma_i^2 + \underline{\sigma}^2}}\right) \\
&= \sum_{i=1}^{n}\sqrt{\gamma_i^2 + \underline{\sigma}^2} \ . \quad (41)
\end{aligned}
$$

**Proposition 18.** $Z \mapsto \min_{S \succ 0} \frac{1}{2}\operatorname{Tr}[Z^\top S^{-1} Z] + \frac{1}{2}\operatorname{Tr}(S) + \frac{\underline{\sigma}^2}{2}\operatorname{Tr}(S^{-1})$ is a $\underline{\sigma}$-smooth approximation of $\|\cdot\|_{\mathscr{S},1}$ with parameters $(1, n, 0)$. More precicely: $Z \mapsto \min_{S \succ 0} \frac{1}{2}\operatorname{Tr}[Z^\top S^{-1} Z] + \frac{1}{2}\operatorname{Tr}(S) + \frac{\underline{\sigma}^2}{2}\operatorname{Tr}(S^{-1})$ has a gradient $\underline{\sigma}$-Lipschitz and

$$
0 \le \min_{S \succ 0} \frac{1}{2}\operatorname{Tr}[Z^\top S^{-1} Z] + \frac{1}{2}\operatorname{Tr}(S) + \frac{\underline{\sigma}^2}{2}\operatorname{Tr}(S^{-1}) - \|Z\|_{\mathscr{S},1} = \underline{\sigma}\sum_i \frac{1}{\sqrt{1 + \frac{\gamma_i^2}{\underline{\sigma}^2}} + \frac{\gamma_i}{\underline{\sigma}}} \le \underline{\sigma}n \ . \quad (42)
$$

*Proof.* $\sum_{i=1}^{n \wedge q} \sqrt{\gamma_i^2 + \underline{\sigma}^2}$ is a $\underline{\sigma}$-smooth approximation of $\sum_{i=1}^{n \wedge q} \sqrt{\gamma_i^2} = \|Z\|_{\mathscr{S},1}$, see Beck and Teboulle (2012, Example 4.6).

$$\min_{S \succ 0} \frac{1}{2} \operatorname{Tr}[Z^\top S^{-1} Z] + \frac{1}{2} \operatorname{Tr}(S) + \frac{\underline{\sigma}^2}{2} \operatorname{Tr}(S^{-1}) - \|Z\|_{\mathscr{S},1} = \sum_{i=1}^n \left( \sqrt{\gamma_i^2 + \underline{\sigma}^2} - \gamma_i \right)$$

$$= \sum_{i=1}^n \frac{\underline{\sigma}^2}{\sqrt{\gamma_i^2 + \underline{\sigma}^2} + \gamma_i}$$

$$= \underline{\sigma} \sum_{i=1}^n \frac{1}{\sqrt{1 + \frac{\gamma_i^2}{\underline{\sigma}^2}} + \frac{\gamma_i}{\underline{\sigma}}} \qquad (43)$$

$$\le \underline{\sigma} n . \qquad (44)$$

Moreover this bound is attained when $Z = 0$. $\qquad \square$

It can be shown that with a fixed Lipschitz constant, the proposed smoothing is (at least) a twice better approximation. This can be quantified even more precisely:

**Proposition 19.**

$$0 \le \underbrace{\left( \|\cdot\|_{\mathscr{S},1} \square \omega_{\underline{\sigma}} \right)(Z) - \|Z\|_{\mathscr{S},1}}_{\operatorname{Err}_1(Z)} \le \frac{1}{2} \left( \underbrace{\min_{S \succ 0} \frac{1}{2} \operatorname{Tr}[Z^\top S^{-1} Z] + \frac{1}{2} \operatorname{Tr}(S) + \frac{\underline{\sigma}^2}{2} \operatorname{Tr}(S^{-1}) - \|Z\|_{\mathscr{S},1}}_{\operatorname{Err}_2(Z)} \right) .$$

$$(45)$$

More precisely

$$\frac{1}{2} \operatorname{Err}_2(Z) - \operatorname{Err}_1(Z) = \frac{\underline{\sigma}}{2} \sum_{\gamma_i \ge \underline{\sigma}} \underbrace{\left( \sqrt{1 + \frac{\gamma_i^2}{\underline{\sigma}^2}} - \frac{\gamma_i}{\underline{\sigma}} \right)}_{\ge 0} + \frac{\underline{\sigma}}{2} \sum_{\gamma_i < \underline{\sigma}} \underbrace{\left( \frac{1}{\sqrt{1 + \frac{\gamma_i^2}{\underline{\sigma}^2}} + \frac{\gamma_i}{\underline{\sigma}}} - (1 + \frac{\gamma_i}{\underline{\sigma}})^2 \right)}_{\ge 0} ,$$

$$(46)$$

which means that for a fixed smoothing constant $\underline{\sigma}$, our smoothing is at least twice uniformly better. Moreover the proposed smoothing can be much better, in particular when a lot a singular values are around $\underline{\sigma}$.

*Proof.* Using the formulas of $\operatorname{Err}_1$ (Equation (36)) and $\operatorname{Err}_2$ (Equation (43)), Equation (46) is direct. In Equation (46) the positivity of the first sum is trivial, the positivity of the second can be obtained with an easy function study. $\qquad \square$

### A.6 Schatten 1-norm (nuclear/trace norm) with repetitions

Let $Z^{(1)}, \dots, Z^{(r)}$ be matrices in $\mathbb{R}^{n \times q}$, then we define $Z \in \mathbb{R}^{n \times qr}$ by $Z = [Z^{(1)} | \dots | Z^{(r)}]$.

**Proposition 20.** For the choice $\omega(Z) = \frac{1}{2} \|Z\|^2 + \frac{n \wedge qr}{2}$, then the following holds true:

$$\left( \|\cdot\|_{\mathscr{S},1} \square \omega_{\underline{\sigma}}(\cdot) \right)(Z) = \min_{S \succeq \underline{\sigma} \operatorname{Id}_n} \frac{1}{2} \sum_{l=1}^r \operatorname{Tr} \left( Z^{(l)\top} S^{-1} Z^{(l)} \right) + \frac{1}{2} \operatorname{Tr}(S) . \qquad (47)$$

*Proof.* The result is a direct application of Proposition 4, with $Z = [Z^{(1)} | \dots | Z^{(r)}]$. It suffices to notice that $\operatorname{Tr} Z^\top S^{-1} Z = \sum_{l=1}^r \operatorname{Tr} \left( Z^{(l)\top} S^{-1} Z^{(l)} \right)$. $\qquad \square$

## A.7 Schatten 2-norm (Frobenius norm)

**Proposition 21.** For the choice $\omega(\cdot) = \frac{1}{2}\|\cdot\|^2 + \frac{1}{2}$, and for $Z \in \mathbb{R}^{n \times q}$ then the following holds true:

$$\left(\|\cdot\| \,\square\, \omega_{\underline{\sigma}}\right)(Z) = \min_{\sigma \geq \underline{\sigma}} \left(\frac{1}{2\sigma}\|Z\|^2 + \frac{\sigma}{2}\right) = \begin{cases} \frac{\|Z\|^2}{2\underline{\sigma}} + \frac{\underline{\sigma}}{2} , & \text{if } \|Z\| \leq \underline{\sigma} , \\ \|Z\| , & \text{if } \|Z\| > \underline{\sigma} . \end{cases} \tag{48}$$

*Proof.* Let us recall that $\|\cdot\| = \|\cdot\|_{\mathscr{S},2}$. Therefore

$$\Pi_{\mathcal{B}_{\mathscr{S},2}}\left(\frac{Z}{\underline{\sigma}}\right) = \begin{cases} 0 , & \text{if } \|Z\| \leq \underline{\sigma} , \\ \frac{Z}{\|Z\|} , & \text{if } \|Z\| > \underline{\sigma} . \end{cases} \tag{49}$$

By combining Equation (49) and Lemma 13 with $p^* = 2$, and $c = \frac{1}{2}$, the later yields

$$\left(\|\cdot\| \,\square\, \omega_{\underline{\sigma}}\right)(Z) = \begin{cases} \frac{1}{2\underline{\sigma}}\|Z\|^2 + \frac{\underline{\sigma}}{2} , & \text{if } \|Z\| \leq \underline{\sigma} , \\ \|Z\| , & \text{if } \|Z\| > \underline{\sigma} . \end{cases}$$

$\square$

## A.8 Schatten infinity-norm (spectral norm)

**Proposition 22.** For the choice $\omega(\cdot) = \frac{1}{2}\|\cdot\|^2 + \frac{1}{2}$ and for $Z \in \mathbb{R}^{n \times q}$, then the following holds true:

$$\left(\|\cdot\|_{\mathscr{S},\infty} \,\square\, \omega_{\underline{\sigma}}\right)(Z) = \begin{cases} \frac{1}{2\underline{\sigma}}\|Z\|^2 + \frac{\underline{\sigma}}{2} , & \text{if } \|Z\|_{\mathscr{S},1} \leq \underline{\sigma} , \\ \frac{\underline{\sigma}}{2} \sum_{i=1}^{n \wedge q} \left(\frac{\gamma_i^2}{\underline{\sigma}^2} - \nu^2\right)_+ + \frac{\underline{\sigma}}{2} , & \text{if } \|Z\|_{\mathscr{S},1} > \underline{\sigma} , \end{cases}$$

where $\nu \geq 0$ is defined by the implicit equation

$$\left\|\left(\mathrm{ST}\left(\frac{\gamma_1}{\underline{\sigma}}, \nu\right), \ldots, \mathrm{ST}\left(\frac{\gamma_{n \wedge q}}{\underline{\sigma}}, \nu\right)\right)\right\|_1 = 1 . \tag{50}$$

*Proof.* We remind that $\Pi_{\mathcal{B}_{\mathscr{S},1}}$, the projection over $\mathcal{B}_{\mathscr{S},1}$, is given by Beck (2017, Example 7.31):

$$\Pi_{\mathcal{B}_{\mathscr{S},1}}\left(\frac{Z}{\underline{\sigma}}\right) = \begin{cases} \frac{Z}{\underline{\sigma}} , & \text{if } \|Z\|_{\mathscr{S},1} \leq \underline{\sigma} , \\ V \,\mathrm{diag}(\mathrm{ST}(\frac{\gamma_i}{\underline{\sigma}}, \nu))W^\top , & \text{if } \|Z\|_{\mathscr{S},1} > \underline{\sigma} , \end{cases} \tag{51}$$

$\gamma$ being defined by the implicit equation

$$\left\|\left(\mathrm{ST}\left(\frac{\gamma_1}{\underline{\sigma}}, \nu\right), \ldots, \mathrm{ST}\left(\frac{\gamma_{n \wedge q}}{\underline{\sigma}}, \nu\right)\right)\right\|_1 = 1 . \tag{52}$$

By combining Equation (51) and Lemma 13 (with $p^* = 1, c = \frac{1}{2}$) it follows that

$$(\|\cdot\| \,\square\, \omega_{\underline{\sigma}})(Z) = \begin{cases} \frac{1}{2\underline{\sigma}}\|Z\|^2 + \frac{\underline{\sigma}}{2} , & \text{if } \|Z\|_{\mathscr{S},1} \leq \underline{\sigma} , \\ \frac{1}{2\underline{\sigma}}\|Z\|^2 + \frac{\underline{\sigma}}{2} - \frac{\underline{\sigma}}{2}\left\|\Pi_{\mathcal{B}_{\mathscr{S},1}}\left(\frac{Z}{\underline{\sigma}}\right) - \frac{Z}{\underline{\sigma}}\right\|^2 , & \text{if } \|Z\|_{\mathscr{S},1} > \underline{\sigma} . \end{cases} \tag{53}$$

Let us compute $\left\|\Pi_{\mathcal{B}_{\mathscr{S},1}}\left(\frac{Z}{\underline{\sigma}}\right) - \frac{Z}{\underline{\sigma}}\right\|^2$. If $\|Z\|_{\mathscr{S},1} > \underline{\sigma}$ we have

$$\left\|\Pi_{\mathcal{B}_{\mathscr{S},1}}\left(\frac{Z}{\underline{\sigma}}\right) - \frac{Z}{\underline{\sigma}}\right\|^2 = \frac{1}{\underline{\sigma}^2}\left\|V \,\mathrm{diag}((\gamma_i - \nu\underline{\sigma})_+ - \gamma_i)W^\top\right\|^2 \qquad \text{(using Equation (51))}$$

$$= \frac{1}{\underline{\sigma}^2} \sum_{i=1}^{n \wedge q} \left((\gamma_i - \nu\underline{\sigma})_+ - \gamma_i\right)^2$$

$$= \frac{1}{\underline{\sigma}^2}\left(\sum_{\gamma_i \geq \nu\underline{\sigma}} \nu^2\underline{\sigma}^2 + \sum_{\gamma_i < \nu\underline{\sigma}} \gamma_i^2\right) . \tag{54}$$

By plugging Equation (54) into Equation (53) it follows, that if $\|Z\|_{\mathscr{S},1} > \underline{\sigma}$:

$$\left(\|\cdot\|_{\mathscr{S},\infty} \square \omega_{\underline{\sigma}}\right)(Z) = \frac{1}{2\underline{\sigma}} \sum_{i=1}^{n \wedge q} \gamma_i^2 + \frac{\sigma}{2} - \frac{1}{2\underline{\sigma}} \sum_{\gamma_i \geq \nu \underline{\sigma}}^{n \wedge q} \nu^2 \underline{\sigma}^2 - \frac{1}{2\underline{\sigma}} \sum_{\gamma_i < \nu \underline{\sigma}}^{n \wedge q} \gamma_i^2$$

$$= \frac{1}{2\underline{\sigma}} \sum_{\gamma_i \geq \nu \underline{\sigma}}^{n \wedge q} \left(\gamma_i^2 - \nu^2 \underline{\sigma}^2\right) + \frac{\sigma}{2}$$

$$= \frac{\sigma}{2} \sum_{i=1}^{n \wedge q} \left(\frac{\gamma_i^2}{\underline{\sigma}^2} - \nu^2\right)_+ + \frac{\sigma}{2} \ . \tag{55}$$

Proposition 22 follows by plugging Equation (55) for the case $\|Z\|_{\mathscr{S},1} > \underline{\sigma}$, and the fact that when $\|Z\|_{\mathscr{S},1} \leq \underline{\sigma}$ the result is straightforward. □

*Remark* 23. Since $\nu \mapsto \left\|\left(\mathrm{ST}\left(\frac{\gamma_1}{\underline{\sigma}}, \nu\right), \ldots, \mathrm{ST}\left(\frac{\gamma_{n \wedge q}}{\underline{\sigma}}, \nu\right)\right)\right\|_1$ is decreasing and piecewise linear, the solution of Equation (50) can be computed exactly in $\mathcal{O}(n \wedge q \ \log(n \wedge q))$ operations.

# B  Proofs CLaR

## B.1  Proof of Proposition 6

**Proposition 6** (Proof in Appendix B.1). Any solution of the CLaR Problem (2), $(\hat{\mathrm{B}}, \hat{S}) = (\hat{\mathrm{B}}^{\mathrm{CLaR}}, \hat{S}^{\mathrm{CLaR}})$ is also a solution of:

$$\hat{\mathrm{B}} = \underset{\mathrm{B} \in \mathbb{R}^{p \times q}}{\arg\min} \left(\|\cdot\|_{\mathscr{S},1} \square \omega_{\underline{\sigma}}\right)(Z) + \lambda n \|\mathrm{B}\|_{2,1}$$

$$\hat{S} = \mathrm{ClSqrt}\left(\frac{1}{rq} \hat{R}\hat{R}^{\top}, \underline{\sigma}\right) \ , \text{ where } \hat{R} = [Y^{(1)} - X\hat{\mathrm{B}}| \ldots |Y^{(r)} - X\hat{\mathrm{B}}] \ .$$

*Proof.* Proposition 6 follows from Appendix A.6 by choosing $Z = \frac{1}{\sqrt{rq}}[Y^{(1)} - X\mathrm{B}, \ldots, Y^{(r)} - X\mathrm{B}]$ and by taking the $\arg\min$ over B. □

## B.2  Proof of Proposition 7

**Proposition 7** (Proof in Appendix B.2). CLaR is jointly convex in $(\mathrm{B}, S)$. Moreover, $f$ is convex and smooth on the feasible set, and $\|\cdot\|_{2,1}$ is convex and separable in $\mathrm{B}_{j:}$'s, thus minimizing the objective alternatively in $S$ and in $\mathrm{B}_{j:}$'s (see Algorithm 1) converges to a global minimum.

*Proof.*

$$f(\mathrm{B}, S) = \frac{1}{2nqr} \sum_{1}^{r} \left\|Y^{(l)} - X\mathrm{B}\right\|_{S^{-1}}^2 + \frac{1}{2n} \mathrm{Tr}(S) = \mathrm{Tr}(Z^T S^{-1} Z) + \frac{1}{2n} \mathrm{Tr}(S) \ ,$$

with $Z = \frac{1}{\sqrt{2nqr}}[Y^{(1)} - X\mathrm{B}| \ldots |Y^{(r)} - X\mathrm{B}]$.

First note that the (joint) function $(Z, \Sigma) \mapsto \mathrm{Tr}\, Z^{\top}\Sigma^{-1}Z$ is jointly convex over $\mathbb{R}^{n \times q} \times \mathcal{S}_{++}^n$, see Boyd and Vandenberghe (2004, Example 3.4). This means that $f$ is jointly convex in $(Z, S)$, moreover $\mathrm{B} \mapsto \frac{1}{\sqrt{2nqr}}[Y^{(1)} - X\mathrm{B}| \ldots |Y^{(r)} - X\mathrm{B}]$ is linear in B, thus $f$ is jointly convex in $(\mathrm{B}, S)$, meaning that $(\mathrm{B}, S) \to f + \lambda \|\cdot\|_{2,1}$ is jointly convex in $(\mathrm{B}, S)$. Moreover the constraint set is convex and thus solving CLaR is a convex problem.

The function $f$ is convex and smooth on the feasible set and $\|\cdot\|_{2,1}$ is convex in B and separable in $\mathrm{B}_{j:}$'s, thus (see Tseng 2001; Tseng and Yun 2009) $f + \lambda \|\cdot\|_{2,1}$ can be minimized through coordinate descent in $S$ and the $\mathrm{B}_{j:}$'s (on the feasible set). □

## B.3  Proof of Proposition 8

**Proposition 8** (Minimization in $S$; proof in Appendix B.3). *Let* $B \in \mathbb{R}^{n \times q}$ *be fixed. The minimization of* $f(B, S)$ *w.r.t.* $S$ *with the constraint* $S \succeq \underline{\sigma} \operatorname{Id}_n$ *admits the closed-form solution:*

$$S = \operatorname{ClSqrt}\left(\frac{1}{rq}\sum_{l=1}^{r}(Y^{(l)} - XB)(Y^{(l)} - XB)^\top, \underline{\sigma}\right) . \tag{12}$$

*Proof.* Minimizing $f(B, \cdot)$ amounts to solving

$$\underset{S \succeq \underline{\sigma} \operatorname{Id}_n}{\arg\min} \tfrac{1}{2}\|Z\|_{S^{-1}}^2 + \tfrac{1}{2}\operatorname{Tr}(S) , \quad \text{with } Z = \frac{1}{\sqrt{r}}[Z^{(1)}|\dots|Z^{(l)}] . \tag{56}$$

The solution is $\operatorname{ClSqrt}\left(ZZ^\top, \underline{\sigma}\right)$ (see Massias et al. 2018a, Appendix A2), and $ZZ^\top = \frac{1}{r}\sum_{l=1}^{r}Z^{(l)}Z^{(l)\top}$.  $\square$

## B.4  Proof of Proposition 9

**Proposition 9** (Proof in Appendix B.4). *For a fixed* $S \in \mathcal{S}_{++}^n$, *each step of the block minimization of* $f(\cdot, S) + \lambda \|\cdot\|_{2,1}$ *in the* $j^{th}$ *line of* B *admits a closed-form solution:*

$$B_{j:} = \operatorname{BST}\left(B_{j:} + \frac{X_{:j}^\top S^{-1}(\bar{Y} - XB)}{\|X_{:j}\|_{S^{-1}}^2}, \frac{\lambda nq}{\|X_{:j}\|_{S^{-1}}^2}\right) . \tag{13}$$

*Proof.* The function to minimize is the sum of a smooth term $f(\cdot, S)$ and a non-smooth but separable term, $\|\cdot\|_{2,1}$, whose proximal operator [6] can be computed:

- $f$ is $\|X_{:j}\|_{S^{-1}}^2 /nq$-smooth with respect to $B_{j:}$, with partial gradient $\nabla_j f(\cdot, S) = -\frac{1}{nq}X_{:j}^\top S^{-1}(\bar{Y} - XB)$,

- $\|B\|_{2,1} = \sum_{j=1}^{p}\|B_{j:}\|$ is row-wise separable over B, with $\operatorname{prox}_{\lambda nq/\|X_{:j}\|_{S^{-1}}^2, \|\cdot\|}(\cdot) = \operatorname{BST}\left(\cdot, \frac{\lambda nq}{\|X_{:j}\|_{S^{-1}}^2}\right).$

Hence, proximal block-coordinate descent converges (Tseng and Yun, 2009), and the update are given by Equation (13). The closed-form formula arises since the smooth part of the objective is quadratic and isotropic *w.r.t.* $B_{j:}$.  $\square$

## B.5  Proof of $\lambda_{\max}$ CLaR

*Proof.* First notice that if $\hat{B} = 0$, then $\hat{S} = \operatorname{ClSqrt}\left(\frac{1}{qr}\sum_{l=1}^{r}Y^{(l)}Y^{(l)\top}, \underline{\sigma}\right) \triangleq S_{\max}$ .

Fermat's rules states that

$$\begin{aligned}
\hat{B} = 0 &\Leftrightarrow 0 \in \partial\big(f(\cdot, S_{\max}) + \lambda\|\cdot\|_{2,1}\big)(0) \\
&\Leftrightarrow -\nabla f(\cdot, S_{\max}) \in \lambda \mathcal{B}_{\|\cdot\|_{2,\infty}} \\
&\Leftrightarrow \frac{1}{nq}\left\|X^\top S_{\max}^{-1}\bar{Y}\right\|_{2,\infty} \triangleq \lambda_{\max} \leq \lambda .
\end{aligned} \tag{57}$$

$\square$

## B.6 Proof of dual formulation

**Proposition 24.** With $\hat{\Theta} = (\hat{\Theta}^{(1)}, \ldots, \hat{\Theta}^{(r)})$, the dual formulation of [Problem (2)] is

$$\hat{\Theta} = \underset{(\Theta^{(1)}, \ldots, \Theta^{(r)}) \in \Delta_{X,\lambda}}{\arg\max} \frac{\sigma}{2} \left( 1 - \frac{qn\lambda^2}{r} \sum_{l=1}^{r} \operatorname{Tr} \Theta^{(l)} \Theta^{(l)\top} \right) + \frac{\lambda}{r} \sum_{l=1}^{r} \left\langle \Theta^{(l)}, Y^{(l)} \right\rangle \ ,$$

with $\bar{\Theta} = \frac{1}{r} \sum_1^r \Theta^{(l)}$ and

$$\Delta_{X,\lambda} = \left\{ (\Theta^{(1)}, \ldots, \Theta^{(r)}) \in (\mathbb{R}^{n \times q})^r : \left\| X^\top \bar{\Theta} \right\|_{2,\infty} \leq 1, \left\| \sum_{l=1}^{r} \Theta^{(l)} \Theta^{(l)\top} \right\|_2 \leq \frac{r}{\lambda^2 n^2 q} \right\} \ .$$

In [Algorithm 1] the dual point $\Theta$ at iteration $t$ is obtained through a residual rescaling similar to the way the dual point is created, *i.e.,* $\Theta^{(l)} = \frac{1}{nq\lambda}(Y^{(l)} - X\mathrm{B})$ (with B the current primal iterate); then the dual point hence created is projected on $\Delta_{X,\lambda}$ .

*Proof.* Let the primal optimum be

$$p^* \triangleq \min_{\substack{\mathrm{B} \in \mathbb{R}^{p \times q} \\ S \succeq \underline{\sigma} \operatorname{Id}_n}} \frac{1}{2nqr} \sum_{l=1}^{r} \|Y^{(l)} - X\mathrm{B}\|_{S^{-1}}^2 + \frac{1}{2n} \operatorname{Tr}(S) + \lambda \|\mathrm{B}\|_{2,1}$$

Then

$$p^* = \min_{\substack{\mathrm{B} \in \mathbb{R}^{p \times q} \\ R^{(l)} = Y^{(l)} - X\mathrm{B}, \ \forall l \in [r] \\ S \succeq \underline{\sigma} \operatorname{Id}_n}} \frac{1}{2nqr} \sum_{l=1}^{r} \|R^{(l)}\|_{S^{-1}}^2 + \frac{1}{2n} \operatorname{Tr}(S) + \lambda \|\mathrm{B}\|_{2,1}$$

$$= \min_{\substack{\mathrm{B} \in \mathbb{R}^{p \times q} \\ R^{(1)}, \ldots, R^{(r)} \in \mathbb{R}^{n \times q} \\ S \succeq \underline{\sigma} \operatorname{Id}_n}} \max_{\Theta^{(1)}, \ldots, \Theta^{(r)} \in \mathbb{R}^{n \times q}} \frac{1}{2nqr} \sum_{l=1}^{r} \|R^{(l)}\|_{S^{-1}}^2 + \frac{1}{2n} \operatorname{Tr}(S)$$

$$+ \lambda \|\mathrm{B}\|_{2,1} + \frac{\lambda}{r} \sum_{l=1}^{r} \left\langle \Theta^{(l)}, Y^{(l)} - X\mathrm{B} - R^{(l)} \right\rangle \ .$$

Since Slater's conditions are met min and max can be inverted:

$$p^* = \max_{\Theta^{(1)}, \ldots, \Theta^{(r)} \in \mathbb{R}^{n \times q}} \min_{\substack{\mathrm{B} \in \mathbb{R}^{p \times q} \\ R^{(1)}, \ldots, R^{(r)} \in \mathbb{R}^{n \times q} \\ S \succeq \underline{\sigma} \operatorname{Id}_n}} \frac{1}{2nqr} \sum_{l=1}^{r} \|R^{(l)}\|_{S^{-1}}^2 + \frac{1}{2n} \operatorname{Tr}(S) \tag{58}$$

$$+ \lambda \|\mathrm{B}\|_{2,1} + \frac{\lambda}{r} \sum_{l=1}^{r} \left\langle \Theta^{(l)}, Y^{(l)} - X\mathrm{B} - R^{(l)} \right\rangle$$

$$= \max_{\Theta^{(1)}, \ldots, \Theta^{(r)} \in \mathbb{R}^{n \times q}} \left( \min_{S \succeq \underline{\sigma} \operatorname{Id}_n} \frac{1}{r} \sum_{l=1}^{r} \min_{R^{(l)} \in \mathbb{R}^{n \times q}} \left( \frac{\|R^{(l)}\|_{S^{-1}}^2}{2nq} - \left\langle \Theta^{(l)}, R^{(l)} \right\rangle \right) + \frac{1}{2n} \operatorname{Tr}(S) \right.$$

$$\left. + \lambda \min_{\mathrm{B} \in \mathbb{R}^{p \times q}} \left( \|\mathrm{B}\|_{2,1} - \left\langle \bar{\Theta}, X\mathrm{B} \right\rangle \right) + \frac{\lambda}{r} \sum_{l=1}^{r} \left\langle \Theta^{(l)}, Y^{(l)} \right\rangle \right) \ . \tag{59}$$

Morover we have

$$\min_{R^{(l)} \in \mathbb{R}^{n \times q}} \left( \frac{\|R^{(l)}\|_{S^{-1}}^2}{2nq} - \left\langle \Theta^{(l)}, R^{(l)} \right\rangle \right) = -\frac{nq\lambda^2}{2} \left\langle \Theta^{(l)} \Theta^{(l)\top}, S \right\rangle$$

and

$$\min_{\mathrm{B}\in\mathbb{R}^{p\times q}} \left( \|\mathrm{B}\|_{2,1} - \langle \bar{\Theta}, X\mathrm{B}\rangle \right) = -\max \left( \langle X^{\top}\bar{\Theta}, \mathrm{B}\rangle - \|\mathrm{B}\|_{2,1} \right) = -\iota_{\mathcal{B}_{2,\infty}}(X^{\top}\bar{\Theta}) \ .$$

This leads to:

$$d^* = \max_{\Theta^{(1)},\dots,\Theta^{(r)}\in\mathbb{R}^{n\times q}} \min_{S\succeq\underline{\sigma}\,\mathrm{Id}_n} -\frac{1}{r}\sum_{l=1}^{r}\frac{nq\lambda^2}{2}\left\langle \Theta^{(l)}\Theta^{(l)\top}, S\right\rangle - \lambda\iota_{\mathcal{B}_{2,\infty}}(X^{\top}\bar{\Theta}) + \frac{\mathrm{Tr}(S)}{2n}$$

$$+ \frac{\lambda}{r}\sum_{l=1}^{r}\langle\Theta^{(l)}, Y^{(l)}\rangle$$

$$= \max_{\Theta^{(1)},\dots,\Theta^{(r)}\in\mathbb{R}^{n\times q}} \frac{1}{2n}\min_{S\succeq\underline{\sigma}\,\mathrm{Id}_n}\left( \left\langle \mathrm{Id}_n, S\right\rangle - \frac{qn^2\lambda^2}{r}\sum_{l=1}^{r}\left\langle \Theta^{(l)}\Theta^{(l)\top}, S\right\rangle \right) - \lambda\iota_{\mathcal{B}_{2,\infty}}(X^{\top}\bar{\Theta})$$

$$+ \frac{\lambda}{r}\sum_{l=1}^{r}\langle\Theta^{(l)}, Y^{(l)}\rangle$$

$$= \max_{\Theta^{(1)},\dots,\Theta^{(r)}\in\mathbb{R}^{n\times q}} \frac{1}{2n}\min_{S\succeq\underline{\sigma}\,\mathrm{Id}_n}\left\langle \mathrm{Id}_n - \frac{qn^2\lambda^2}{r}\sum_{l=1}^{r}\Theta^{(l)}\Theta^{(l)\top}, S\right\rangle - \lambda\iota_{\mathcal{B}_{2,\infty}}(X^{\top}\bar{\Theta})$$

$$+ \frac{\lambda}{r}\sum_{l=1}^{r}\langle\Theta^{(l)}, Y^{(l)}\rangle \ . \tag{60}$$

$$\min_{S\succeq\underline{\sigma}\,\mathrm{Id}_n}\left\langle \mathrm{Id}_n - \frac{qn^2\lambda^2}{r}\sum_{l=1}^{r}\Theta^{(l)}\Theta^{(l)\top}, S\right\rangle$$

$$= \begin{cases} \left\langle \mathrm{Id}_n - \frac{qn^2\lambda^2}{r}\sum_{l=1}^{r}\Theta^{(l)}\Theta^{(l)\top}, \underline{\sigma}\right\rangle \ , & \text{if } \mathrm{Id}_n \succeq \frac{qn^2\lambda^2}{r}\sum_{l=1}^{r}\Theta^{(l)}\Theta^{(l)\top} \ , \\ -\infty \ , & \text{otherwise.} \end{cases} \tag{61}$$

It follows that the dual problem of CLaR is

$$\max_{(\Theta^{(1)},\dots,\Theta^{(r)})\in\Delta_{X,\lambda}} \frac{\underline{\sigma}}{2}\left( 1 - \frac{qn\lambda^2}{r}\sum_{l=1}^{r}\mathrm{Tr}\,\Theta^{(l)}\Theta^{(l)\top} \right) + \frac{\lambda}{r}\sum_{l=1}^{r}\left\langle\Theta^{(l)}, Y^{(l)}\right\rangle \ , \tag{62}$$

where $\Delta_{X,\lambda} \triangleq \left\{ (\Theta^{(1)},\dots,\Theta^{(r)}) \in \mathbb{R}^{n\times q\times r} : \|X^{\top}\bar{\Theta}\|_{2,\infty} \le 1, \|\sum_{l=1}^{r}\Theta^{(l)}\Theta^{(l)\top}\|_2 \le \frac{r}{\lambda^2 n^2 q} \right\}$.
$\square$

## B.7 Proof of Remark 11

*Remark* 11. Once $\mathrm{cov}_Y \triangleq \frac{1}{r}\sum_1^r Y^{(l)}Y^{(l)\top}$ is pre-computed, the cost of updating $S$ does not depend on $r$, *i.e.,* is the same as working with averaged data. Indeed, with $R = [Y^{(1)} - X\mathrm{B}|\dots|Y^{(r)} - X\mathrm{B}]$, the following computation can be done in $\mathcal{O}(qn^2)$ (details are in Appendix B.7).

$$RR^{\top} = \mathrm{RRT}(\mathrm{cov}_Y, Y, X, \mathrm{B}) \triangleq r\mathrm{cov}_Y + r(X\mathrm{B})(X\mathrm{B})^{\top} - r\bar{Y}^{\top}(X\mathrm{B}) - r(X\mathrm{B})^{\top}\bar{Y} \ . \tag{15}$$

*Proof.*

$$RR^{\top} = \sum_{l=1}^{r}R^{(l)}R^{(l)\top}$$

$$= \sum_{l=1}^{r}(Y^{(l)} - X\mathrm{B})(Y^{(l)} - X\mathrm{B})^{\top}$$

$$= \sum_{l=1}^{r}Y^{(l)}Y^{(l)\top} - \sum_{1}^{r}Y^{(l)}(X\mathrm{B})^{\top} - \sum_{1}^{r}X\mathrm{B}Y^{(l)\top} + rX\mathrm{B}(X\mathrm{B})^{\top}$$

$$= r\mathrm{cov}_Y - r\bar{Y}^{\top}X\mathrm{B} - r(X\mathrm{B})^{\top}\bar{Y} + rX\mathrm{B}(X\mathrm{B})^{\top} \tag{63}$$

$\square$

**B.8 Statistical comparison**

In this subsection, we show the statistical interest of using all repetitions of the experiments instead of using a mere averaging as SGCL would do (remind that the later is equivalent to CLaR with $r = 1$ and $Y^{(1)} = \bar{Y}$, see Remark 3).

Let us introduce $\Sigma^*$, the true covariance matrix of the noise (*i.e.,* $\Sigma^* = S^{*2}$ with our notation). In SGCL and CLaR alternate minimization consists in a succession of estimations of $B^*$ and $\Sigma^*$ (more precisely $S = \mathrm{ClSqrt}(\Sigma, \underline{\sigma})$ is estimated along the process). In this section we explain why the estimation of $\Sigma^*$ provided by CLaR has better statistical properties than that of SGCL. For that, we can compare the estimates of $\Sigma^*$ one would obtain provided that the true parameter $B^*$ is known by both SGCL and CLaR. In such "ideal" scenario, the associated estimators of $\Sigma^*$ could be written:

$$\hat{\Sigma}^{\mathrm{CLaR}} \triangleq \frac{1}{qr} \sum_{l=1}^{r} (Y^{(l)} - X\hat{B})(Y^{(l)} - X\hat{B})^{\top} \ , \tag{64}$$

$$\hat{\Sigma}^{\mathrm{SGCL}} \triangleq \frac{1}{qr} \Big( \sum_{l=1}^{r} Y^{(l)} - X\hat{B} \Big) \Big( \sum_{l=1}^{r} Y^{(l)} - X\hat{B} \Big)^{\top}, \tag{65}$$

with $\hat{B} = B^*$, and satisfy the following properties:

**Proposition 25.** Provided that the true signal is known, and that the covariance estimator $\hat{\Sigma}^{\mathrm{CLaR}}$ and $\hat{\Sigma}^{\mathrm{SGCL}}$ are defined thanks to Equations (64) and (65), then one can check that

$$\mathbb{E}(\hat{\Sigma}^{\mathrm{CLaR}}) = \mathbb{E}(\hat{\Sigma}^{\mathrm{SGCL}}) = \Sigma^* \ , \tag{66}$$

$$\mathrm{cov}(\hat{\Sigma}^{\mathrm{CLaR}}) = \frac{1}{r} \mathrm{cov}(\hat{\Sigma}^{\mathrm{SGCL}}) \ . \tag{67}$$

Proposition 25 states that $\hat{\Sigma}^{\mathrm{CLaR}}$ and $\hat{\Sigma}^{\mathrm{SGCL}}$ are unbiased estimators of $\Sigma^*$ but our newly introduced CLaR, improves the estimation of the covariance structure by a factor $r$, the number of repetitions performed.

Empirically[7], we have also observed that $\hat{\Sigma}^{\mathrm{CLaR}}$ has larger eigenvalues than $\hat{\Sigma}^{\mathrm{SGCL}}$, leading to a less biased estimation of $S^*$ after clipping the singular values.

Let us recall that

$$\Sigma^{\mathrm{SGCL}} = \frac{1}{qr} \Big( \sum_{l=1}^{r} R^{(l)} \Big) \Big( \sum_{l=1}^{r} R^{(l)} \Big)^{\top} \quad \text{and} \quad \Sigma^{\mathrm{CLaR}} = \frac{1}{qr} \sum_{l=1}^{r} R^{(l)} R^{(l)\top} \ . \tag{68}$$

**Proof of Equation (66)**

*Proof.* If $B = B^*$, $R^{(l)} = S^* E^{(l)}$, where $E^{(l)}$ are random matrices with normal i.i.d. entries, and the result trivially follows. □

**Proof of Equation (67)**

*Proof.* If $\hat{B} = B^*$, $Y^{(l)} - X\hat{B} = S^* E^{(l)}$, where the $E^{(l)}$'s are random matrices with normal i.i.d. entries.

Now, on the one hand :

$$\hat{\Sigma}^{\mathrm{SGCL}} = \frac{1}{qr} \Big( \sum_{l=1}^{r} S^* E^{(l)} \Big) \Big( \sum_{l=1}^{r} S^* E^{(l)} \Big)^{\top} \ .$$

Since $\frac{1}{\sqrt{r}} \sum_{l=1}^{r} S^* E^{(l)} \underset{law}{\sim} S^* E$ it follows that

$$\hat{\Sigma}^{\mathrm{SGCL}} \underset{law}{\sim} \frac{1}{q} S^* E (S^* E)^{\top},$$

$$\mathrm{cov}(\hat{\Sigma}^{\mathrm{SGCL}}) = \frac{1}{q^2} \mathrm{cov}(S^* E (S^* E)^{\top}) \ .$$

On the other hand:

$$\hat{\Sigma}^{\mathrm{CLaR}} = \frac{1}{qr} \sum_{l=1}^{r} S^* \mathrm{E}^{(l)} (S^* \mathrm{E}^{(l)})^\top \ .$$

Since the $\mathrm{E}^{(l)}$'s are independent it follows that

$$\mathrm{cov}(\hat{\Sigma}^{\mathrm{CLaR}}) = \frac{1}{r^2 q^2} \sum_{l=1}^{r} \mathrm{cov}\left( S^* \mathrm{E}^{(l)} (S^* \mathrm{E}^{(l)})^\top \right) = \frac{1}{r^2 q^2} \sum_{l=1}^{r} \mathrm{cov}\left( S^* \mathrm{E}(S^* \mathrm{E})^\top \right)$$

$$= \frac{1}{r q^2} \mathrm{cov}\left( S^* \mathrm{E}(S^* \mathrm{E})^\top \right) = \frac{1}{r} \mathrm{cov}\left( \hat{\Sigma}^{\mathrm{SGCL}} \right) \ .$$

$\square$

# C    Alternative estimators

We compare CLaR to several estimators: SGCL (Massias et al., 2018a), the (smoothed) $\ell_{2,1}$-Maximum Likelihood ($\ell_{2,1}$-MLE), and a version of the $\ell_{2,1}$-MLE with multiple repetitions ($\ell_{2,1}$-MLER), an $\ell_{2,1}$ penalized version of the  Multivariate Regression with Covariance Estimation (Rothman et al., 2010) ($\ell_{2,1}$-MRCE), an $\ell_{2,1}$ penalized version of $\ell_{2,1}$-MRCE with repetitions ($\ell_{2,1}$-MRCER) and the Multi-Task Lasso (Obozinski et al. 2010, MTL). The cost of an epoch of block coordinate descent and the cost of computing the duality gap for each algorithm are summarized in Table 1. The updates of each algorithms are summarized in Table 2.

CLaR solves Problem (2) and SGCL solves Problem (4), let us introduce the definition of the alternative estimation procedures.

## C.1    Multi-Task Lasso (MTL)

The MTL (Obozinski et al., 2010) is the classical estimator used when the additive noise is supposed to be homoscedastic (with no correlation). MTL is obtained by solving:

$$\hat{\mathrm{B}}^{\mathrm{MTL}} \in \argmin_{\mathrm{B} \in \mathbb{R}^{p \times q}} \frac{1}{2nq} \left\| \bar{Y} - X\mathrm{B} \right\|^2 + \lambda \left\| \mathrm{B} \right\|_{2,1} \ . \tag{69}$$

*Remark* 26. It can be seen that trying to use all the repetitions in the MTL leads to MTL itself because $\left\| \bar{Y} - X\mathrm{B} \right\|^2 = \frac{1}{r} \sum_l \left\| Y^{(l)} - X\mathrm{B} \right\|^2$.

## C.2    $\ell_{2,1}$-Maximum Likelihood ($\ell_{2,1}$-MLE)

Here we study a penalized Maximum Likelihood Estimator (Chen and Banerjee, 2017) ($\ell_{2,1}$-MLE). When minimizing $\ell_{2,1}$-Maximum Likelihood the natural parameters of the problem are the regression coefficients B and the precision matrix $\Sigma^{-1}$. Since real M/EEG covariance matrices are not full rank, one has to be algorithmically careful when $\Sigma$ becomes singular. To avoid such numerical errors and to be consistent with the smoothed estimator proposed in the paper (CLaR), let us define the (smoothed) $\ell_{2,1}$-MLE as following:

$$(\hat{\mathrm{B}}^{\ell_{2,1}-\mathrm{MLE}}, \hat{\Sigma}^{\ell_{2,1}-\mathrm{MLE}}) \in \argmin_{\substack{\mathrm{B} \in \mathbb{R}^{p \times q} \\ \Sigma \succeq \underline{\sigma}^2/r^2}} \|\bar{Y} - X\mathrm{B}\|_{\Sigma^{-1}}^2 - \log \det(\Sigma^{-1}) + \lambda \left\| \mathrm{B} \right\|_{2,1} \ , \tag{70}$$

and its repetitions version ($\ell_{2,1}$-MLER):

$$(\hat{\mathrm{B}}^{\ell_{2,1}\mathrm{MLER}}, \hat{\Sigma}^{\ell_{2,1}\mathrm{MLER}}) \in \argmin_{\substack{\mathrm{B} \in \mathbb{R}^{p \times q} \\ \Sigma \succeq \underline{\sigma}^2}} \sum_{1}^{r} \|Y^{(l)} - X\mathrm{B}\|_{\Sigma^{-1}}^2 - \log \det(\Sigma^{-1}) + \lambda \left\| \mathrm{B} \right\|_{2,1} \ . \tag{71}$$

Problems (70) and (71) are not convex because the objective functions are not convex in $(\mathrm{B}, \Sigma^{-1})$, however they are biconvex, *i.e.,* convex in B and convex in $\Sigma^{-1}$. Alternate minimization can be used to solve Problems (70) and (71), but without guarantees to converge toward a global minimum.

**Minimization in** $B_{j:}$   As for CLaR and SGCL the updates in $B_{j:}$'s for $\ell_{2,1}$-MLE and $\ell_{2,1}$-MLER clearly read:

$$B_{j:} = \text{BST}\left(B_{j:} + \frac{X_{:j}^\top \Sigma^{-1}(\bar{Y} - XB)}{\|X_{:j}\|_{\Sigma^{-1}}^2}, \frac{\lambda n q}{\|X_{:j}\|_{\Sigma^{-1}}^2}\right) \ . \tag{72}$$

**Minimization in** $\Sigma^{-1}$**:**   for $\ell_{2,1}$-MLE (*resp. for* $\ell_{2,1}$-MLER) the update in $\Sigma$ reads

$$\Sigma = \text{Cl}(\Sigma^{\text{EMP}}, \underline{\sigma}^2) \quad (\textit{resp. } \Sigma = \text{Cl}(\Sigma^{\text{EMP},r}, \underline{\sigma}^2)) \ , \tag{73}$$

with $\Sigma^{\text{EMP}} \triangleq \frac{1}{q}(\bar{Y} - XB)(\bar{Y} - XB)^\top$ (*resp.* $\Sigma^{\text{EMP},r} \triangleq \frac{1}{rq}\sum_{l=1}^{r}(Y^{(l)} - XB)(Y^{(l)} - XB)^\top$)

Let us prove the last result. Minimizing Problem (70) in $\Sigma^{-1}$ amounts to solving

$$\hat{\Sigma}^{-1} \in \underset{0 \prec \Sigma^{-1} \preceq 1/\underline{\sigma}^2}{\arg\min} \quad \langle \Sigma^{\text{EMP}}, \Sigma^{-1}\rangle - \log\det(\Sigma^{-1}) \ . \tag{74}$$

**Theorem 27.** Let $\Sigma^{\text{EMP}} = U\,\text{diag}(\sigma_i^2)U^\top$ be an eigenvalue decomposition of $\Sigma^{\text{EMP}}$, a solution to Problem (74) is given by:

$$\hat{\Sigma}^{-1} = U\,\text{diag}\left(\frac{1}{\sigma_i^2 \vee \underline{\sigma}^2}\right)U^\top \tag{75}$$

Theorem 27 is very intuitive, the solution of the smoothed optimization problem (74) is the solution of the non-smoothed problem, where the eigenvalues of the solution have been clipped to satisfy the constraint. Let us proove this result.

*Proof.* The KKT conditions of Problem (74) for conic programming (see Boyd and Vandenberghe 2004, p. 267) state that the optimum in the primal $\hat{\Sigma}^{-1}$ and the optimum in the dual $\hat{\Gamma}$ should satisfy:

$$\Sigma^{\text{EMP}} - \hat{\Sigma} + \hat{\Gamma} = 0 \ , \qquad\qquad \hat{\Gamma}^\top\left(\hat{\Sigma}^{-1} - \frac{1}{\underline{\sigma}^2}\,\text{Id}_n\right) = 0 \ ,$$

$$\hat{\Gamma} \in \mathcal{S}_+^n \ , \qquad\qquad 0 \prec \hat{\Sigma}^{-1} \preceq \frac{1}{\underline{\sigma}^2} \ .$$

Since Problem (74) is convex these conditions are also sufficient. Let us propose a primal-dual point $(\hat{\Sigma}^{-1}, \hat{\Gamma})$ satisfying the KKT conditions. Let $\Sigma^{\text{EMP}} = U\,\text{diag}(\sigma_i^2)U^\top$ be an eigenvalue decomposition of $\Sigma^{\text{EMP}}$, one can check that

$$\hat{\Sigma}^{-1} = U\,\text{diag}(\frac{1}{\sigma_i^2 \vee \underline{\sigma}^2})U^\top \ ,$$

$$\hat{\Gamma} = U\,\text{diag}(\sigma_i^2 \vee \underline{\sigma}^2 - \sigma_i^2)U^\top \ .$$

verify the KKT conditions, leading to the desired result. $\qquad\qquad\square$

### C.3   Multivariate Regression with Covariance Estimation  (MRCE)

MRCE (Rothman et al., 2010) jointly estimates the regression coefficients (assumed to be sparse) and the precision matrix (*i.e.,* the inverse of the covariance matrix), which is supposed to be sparse as well. Originally in Rothman et al. (2010) the sparsity enforcing term on the regression coefficients was an $\ell_1$-norm, which is not well suited for our problem, that is why in Appendix C.3.2 we introduce an $\ell_{2,1}$ penalized version of MRCE: $\ell_{2,1}$-MRCE.

### C.3.1   Multivariate Regression with Covariance Estimation

$\ell_{2,1}$-MRCE if defined as the solution of the following optimization problem:

$$(\hat{B}^{\text{MRCE}}, \hat{\Sigma}^{\text{MRCE}}) \in \underset{\substack{B\in\mathbb{R}^{p\times q}\\ \Sigma^{-1}\succ 0}}{\arg\min} \ \left\|\bar{Y} - XB\right\|_{\Sigma^{-1}}^2 - \log\det(\Sigma^{-1}) + \lambda\|B\|_1 + \mu\left\|\Sigma^{-1}\right\|_1 \ . \tag{76}$$

Problem (76) is not convex, but can be solved heuristically (see Rothman et al. 2010 for details) by coordinate descent doing soft-tresholdings for the udpates in $B_{j:}$'s and solving a Graphical Lasso (Friedman et al., 2008) for the update in $\Sigma^{-1}$. The $\ell_1$-norm being not well suited for our problem, we introduce an $\ell_{2,1}$ version of MRCE.

## C.3.2 Multivariate Regression with Covariance Estimation with $l_{2,1}$-norm ($\ell_{2,1}$-MRCE)

The $\ell_1$-norm penalization on the regression penalization B being not well suited for our problem, one can think to an $\ell_{2,1}$-penalized version of MRCE defined as follow:

$$(\hat{B}^{\ell_{2,1}\text{MRCE}}, \hat{\Sigma}^{\ell_{2,1}\text{MRCE}}) \in \underset{\substack{B \in \mathbb{R}^{p \times q} \\ \Sigma^{-1} \succ 0}}{\arg\min} \left\| \bar{Y} - XB \right\|_{\Sigma^{-1}}^2 - \log\det(\Sigma^{-1}) + \lambda \left\| B \right\|_{2,1} + \mu \left\| \Sigma^{-1} \right\|_1 \ . \tag{77}$$

In order to combine $\ell_{2,1}$-MRCE to take take advantage of all the repetitions, one can think of the following estimator:

$$(\hat{B}^{\ell_{2,1}\text{MRCER}}, \hat{\Sigma}^{\ell_{2,1}\text{MRCER}}) \in \underset{\substack{B \in \mathbb{R}^{p \times q} \\ \Sigma^{-1} \succeq 0}}{\arg\min} \sum_1^r \left\| Y^{(l)} - XB \right\|_{\Sigma^{-1}}^2 - \log\det(\Sigma^{-1}) + \lambda \left\| B \right\|_{2,1} + \mu \left\| \Sigma^{-1} \right\|_1 \ . \tag{78}$$

As for Appendix C.3.1, Problem (77) (*resp.* (78)) can be heuristically solved through coordinate descent.

**Update in $B_{j:}$**    It is the same as $\ell_{2,1}$-MLE and $\ell_{2,1}$-MLER:

$$B_{j:} = \text{BST} \left( B_{j:} + \frac{X_{:j}^{\top} \Sigma^{-1} (\bar{Y} - XB)}{\| X_{:j} \|_{\Sigma^{-1}}^2}, \frac{\lambda n q}{\| X_{:j} \|_{\Sigma^{-1}}^2} \right) \ . \tag{79}$$

**Update in $\Sigma^{-1}$**    Minimizing (77) in $\Sigma^{-1}$ amounts to solve:

$$\text{glasso}(\Sigma, \mu) \triangleq \underset{\Sigma^{-1} \succ 0}{\arg\min} \ \langle \Sigma^{\text{EMP}}, \Sigma^{-1} \rangle - \log\det(\Sigma^{-1}) + \mu \left\| \Sigma^{-1} \right\|_1 \ . \tag{80}$$

This is a well known and well studied problem (Friedman et al., 2008) that can be solved through co-ordinate descent. For ourselves we used the `scikit-learn` (Pedregosa et al., 2011) implementation of the Graphical Lasso. Note that applying the Graphical Lasso on very ill conditioned empirical covariance matrix such as $\Sigma^{\text{EMP}}$ is very long. We thus only considered $\ell_{2,1}$-MRCER were the Graphical Lasso is applied on $\Sigma^{\text{EMP},r}$.

## C.4 Algorithms summary

Each estimator, proposed or compared to is based on an optimization problem to solve. Each optimization problem is solve with block coordinate descent, whether there is theoretical guarantees for it to converge toward a global minimum (for convex formulations, CLaR, SGCL and MTL), or not (for non-convex formulations, $\ell_{2,1}$-MLE, $\ell_{2,1}$-MLER, $\ell_{2,1}$-MRCER). The cost for the updates for each algorithm can be found in Table 1. The formula for the updates in $B_{j:}$'s and $S/\Sigma$ for each algorithm can be found in Table 2.

Let $T_{S\ \text{update}}$ be the number of updates of B for one update of $S$ or $\Sigma$.

Table 1 – Algorithms cost in time summary

| | CD epoch cost | convex | dual gap cost |
|---|---|---|---|
| CLaR | $\mathcal{O}(\frac{n^3+qn^2}{T_{S\ \text{update}}} + pn^2 + pnq)$ | yes | $\mathcal{O}(rnq + p)$ |
| SGCL | $\mathcal{O}(\frac{n^3+qn^2}{T_{S\ \text{update}}} + pn^2 + pnq)$ | yes | $\mathcal{O}(nq + p)$ |
| $\ell_{2,1}$-MLER | $\mathcal{O}(\frac{n^3+qn^2}{T_{S\ \text{update}}} + pn^2 + pnq)$ | no | not convex |
| $\ell_{2,1}$-MLE | $\mathcal{O}(\frac{n^3+qn^2}{T_{S\ \text{update}}} + pn^2 + pnq)$ | no | not convex |
| $\ell_{2,1}$-MRCER | $\mathcal{O}(\frac{\mathcal{O}(\text{glasso})}{T_{S\ \text{update}}} + pn^2 + pnq)$ | no | not convex |
| MTL | $\mathcal{O}(npq)$ | yes | $\mathcal{O}(nq + p)$ |

Recalling that $\Sigma^{\text{EMP}} \triangleq \frac{1}{q}(\bar{Y} - XB)(\bar{Y} - XB)^{\top}$ and $\Sigma^{\text{EMP},r} \triangleq \frac{1}{rq} \sum_{l=1}^r (Y^{(l)} - XB)(Y^{(l)} - XB)^{\top}$, a summary of the updates in $S/\Sigma$ and $B_{j:}$'s for each algorithm is given in Table 2.

**Comments on Table 2** The updates in $S/\Sigma$ and $\mathrm{B}_{j:}$'s are given in Table 2. Although the updates may look similar, all the algorithms can lead to very different results, see Figures 6, 9, 11 and 13.

Table 2 – Algorithms updates summary

| | update in $\mathrm{B}_{j:}$ | update in $S/\Sigma$ |
|---|---|---|
| CLaR | $\mathrm{B}_{j:} = \mathrm{BST}\left(\mathrm{B}_{j:} + \frac{X_{:j}^\top S^{-1}(\bar{Y}-X\mathrm{B})}{\|X_{:j}\|_{S^{-1}}^2}, \frac{\lambda nq}{\|X_{:j}\|_{S^{-1}}^2}\right)$ | $S = \mathrm{ClSqrt}(\Sigma^{\mathrm{EMP},r}, \underline{\sigma})$ |
| SGCL | $\mathrm{B}_{j:} = \mathrm{BST}\left(\mathrm{B}_{j:} + \frac{X_{:j}^\top S^{-1}(\bar{Y}-X\mathrm{B})}{\|X_{:j}\|_{S^{-1}}^2}, \frac{\lambda nq}{\|X_{:j}\|_{S^{-1}}^2}\right)$ | $S = \mathrm{ClSqrt}(\Sigma^{\mathrm{EMP}}, \underline{\sigma})$ |
| $\ell_{2,1}$-MLER | $\mathrm{B}_{j:} = \mathrm{BST}\left(\mathrm{B}_{j:} + \frac{X_{:j}^\top \Sigma^{-1}(\bar{Y}-X\mathrm{B})}{\|X_{:j}\|_{\Sigma^{-1}}^2}, \frac{\lambda nq}{\|X_{:j}\|_{\Sigma^{-1}}^2}\right)$ | $\Sigma = \mathrm{Cl}(\Sigma^{\mathrm{EMP},r}, \underline{\sigma}^2)$ |
| $\ell_{2,1}$-MLE | $\mathrm{B}_{j:} = \mathrm{BST}\left(\mathrm{B}_{j:} + \frac{X_{:j}^\top \Sigma^{-1}(\bar{Y}-X\mathrm{B})}{\|X_{:j}\|_{\Sigma^{-1}}^2}, \frac{\lambda nq}{\|X_{:j}\|_{\Sigma^{-1}}^2}\right)$ | $\Sigma = \mathrm{Cl}(\Sigma^{\mathrm{EMP}}, \underline{\sigma}^2)$ |
| $\ell_{2,1}$-MRCER | $\mathrm{B}_{j:} = \mathrm{BST}\left(\mathrm{B}_{j:} + \frac{X_{:j}^\top \Sigma^{-1}(\bar{Y}-X\mathrm{B})}{\|X_{:j}\|_{\Sigma^{-1}}^2}, \frac{\lambda nq}{\|X_{:j}\|_{\Sigma^{-1}}^2}\right)$ | $\Sigma = \mathrm{glasso}(\Sigma^{\mathrm{EMP},r}, \mu)$ |
| MTL | $\mathrm{B}_{j:} = \mathrm{BST}\left(\mathrm{B}_{j:} + \frac{X_{:j}^\top(\bar{Y}-X\mathrm{B})}{\|X_{:j}\|^2}, \frac{\lambda nq}{\|X_{:j}\|^2}\right)$ | no update in $S/\Sigma$ |

# D    Supplementary experiments

In this section we describe the preprocessing pipeline used for the realistic and real data (see Appendix D.1). We then propose time comparison for all the algorithms (see Appendix D.2). And finally we expose supplementary experiments on real data (see Appendix D.3 to D.4).

## D.1    Preprocessing steps for realistic and real data

When using multi-modal data without whitening, one has to rescale properly data, indeed data needs to have the same order of magnitude, otherwise some mode (for example EEG data) could be (almost) completely ignored by the optimization algorithm. The preprocessing pipeline used to rescale realistic data (Figures 4 and 5) and real data (Figures 6, 9, 11 and 13) is described in Algorithm 2.

---

**Algorithm 2** PREPROCESSING STEPS FOR REALISTIC AND REAL DATA

**input :** $X, Y^{(1)}, \ldots, Y^{(r)}$
`// rescale each line of` $X$
**for** $i = 1, \ldots, n$ **do**
  **for** $l = 1, \ldots, r$ **do**
    $Y_{i:}^{(l)} \leftarrow Y_{i:}^{(l)} / \|X_{i:}\|$
  $X_{i:} \leftarrow X_{i:} / \|X_{i:}\|$
`// rescale each column of` $X$
**for** $j = 1, \ldots, q$ **do**
  $X_{:j} \leftarrow X_{:j} / \|X_{:j}\|$
**return** $X, Y^{(1)}, \ldots, Y^{(r)}$

---

## D.2    Time comparison

The goal of this experiment is to show that our algorithm (CLaR) is as costly as a Multi-Task Lasso or other competitors (in the M/EEG context, *i.e.,* $n$ not too large). The time taken by each algorithm to produce Figure 6 (real data, left auditory stimulations) is given in Figure 8. In this experiment the tolerance is set to tol=$10^{-3}$, the safe stopping criterion is duality gap < tol (only available for convex optimization problems). The heuristic stopping criterion is "if the objective do not decrease enough anymore then stop" *i.e.,* if objective$(\mathrm{B}^{(t)}, \Sigma^{(t)})$ − objective$(\mathrm{B}^{(t+1)}, \Sigma^{(t+1)})$ < tol/10 then stop. The safe stopping criterion is only available for CLaR, SGCL and MTL (it takes too much time *i.e.,* more than 10min for SGCL to have a duality gap under the fixed tol, so we remove it).

Figure 8 – *Time comparison, real data, $n = 102$, $p = 7498$, $q = 54$, $r = 56$* Time for each algorithm to produce Figure 6.

**Comment on Figure 8**  Figure 8 shows that if we use the heuristic stopping criterion, CLaR is as fast the other algorithm. In addition CLaR has a safe stopping criterion which only take 2 to 3 more time than the heuristic one (less than 10sec).

### D.3  Supplementary experiments on real data: right auditory stimulations

(a) CLaR  (b) SGCL  (c) $\ell_{2,1}$-MLER  (d) $\ell_{2,1}$-MLE  (e) $\ell_{2,1}$-MRCER  (f) MTL

Figure 9 – *Real data ($n = 102$, $q = 7498$, $q = 76$, $r = 65$)* Sources found in the left hemisphere (top) and the right hemisphere (bottom) after right auditory stimulations.

Figures 9 and 10 show the solution given by each algorithm on real data after right auditory stimulations. As two sources are expected (one in each hemisphere, in bilateral auditory cortices), we vary $\lambda$ by dichotomy between $\lambda_{\max}$ (returning 0 sources) and a $\lambda_{\min}$ (returning more than 2 sources), until finding a lambda giving exactly 2 sources. Figure 9 (*resp.* Figure 10) shows the solution given by the algorithms taking in account all the repetitions (*resp.* only half of the repetitions). When the number of repetitions is high (Figure 9) only CLaR and $\ell_{2,1}$-MLER find one source in each auditory cortex, MTL does find sources only in one hemisphere, all the other algorithms fail by finding sources not

in the auditory cortices at all. Moreover when the number of repetitions is decreasing (Figure 10) $\ell_{2,1}$-MLER fails and only CLaR does find 2 sources, one in each hemisphere. Once again CLaR is more robust and performs better, even when the number of repetitions is low.

(a) CLaR    (b) SGCL    (c) $\ell_{2,1}$-MLER    (d) $\ell_{2,1}$-MLE    (e) $\ell_{2,1}$-MRCER    (f) MTL

Figure 10 – *Real data (n = 102, q = 7498, q = 76, r = 33)* Sources found in the left hemisphere (top) and the right hemisphere (bottom) after right auditory stimulations.

### D.4   Supplementary experiments on real data: left visual stimulations

(a) CLaR    (b) SGCL    (c) $\ell_{2,1}$-MLER    (d) $\ell_{2,1}$-MLE    (e) $\ell_{2,1}$-MRCER    (f) MTL

Figure 11 – *Real data (n = 102, q = 7498, q = 48, r = 71)* Sources found in the left hemisphere (top) and the right hemisphere (bottom) after left visual stimulations.

Figures 11 and 12 show the results for each algorithm after left visual stimulations. As one source is expected (in the right hemisphere), we vary $\lambda$ by dichotomy between $\lambda_{\max}$ (returning 0 sources) and a $\lambda_{\min}$ (returning more than 1 sources), until finding a lambda giving exactly 1 source. When the number of repetitions is high (Figure 11) only CLaR and $\ell_{2,1}$-MLER do find a source in the visual cortex. When the number of repetitions decreases, CLaR and $\ell_{2,1}$-MLER still find one source in the visual cortex, other algorithms fail. This highlights this importance of taking into account the repetitions.

(a) CLaR    (b) SGCL    (c) $\ell_{2,1}$-MLER    (d) $\ell_{2,1}$-MLE    (e) $\ell_{2,1}$-MRCER    (f) MTL

Figure 12 – *Real data (n = 102, q = 7498, q = 48, r = 36)* Sources found in the left hemisphere (top) and the right hemisphere (bottom) after left visual stimulations.

## D.5 Supplementary experiments on real data: right visual stimulations

(a) CLaR     (b) SGCL     (c) $\ell_{2,1}$-MLER     (d) $\ell_{2,1}$-MLE     (e) $\ell_{2,1}$-MRCER     (f) MTL

Figure 13 – *Real data (n = 102, q = 7498, q = 48, r = 61)* Sources found in the left hemisphere (top) and the right hemisphere (bottom) after right visual stimulations.

Figures 13 and 14 show the results for each algorithm after right visual stimulations. As one source is expected (in the left hemisphere), we vary $\lambda$ by dichotomy between $\lambda_{\max}$ (returning 0 sources) and a $\lambda_{\min}$ (returning more than 1 sources), until finding a lambda giving exactly 1 source. When the number of repetitions is high (Figure 13) only CLaR, $\ell_{2,1}$-MLER and MTL do find a source in the visual cortex. When the number of repetitions decreases (Figure 14), only CLaR finds one source in the visual cortex, other algorithms fail. This highlights once again the robustness of CLaR, even with a limited number of repetitions.

(a) CLaR     (b) SGCL     (c) $\ell_{2,1}$-MLER     (d) $\ell_{2,1}$-MLE     (e) $\ell_{2,1}$-MRCER     (f) MTL

Figure 14 – *Real data (n = 102, q = 7498, q = 48, r = 31)* Sources found in the left hemisphere (top) and the right hemisphere (bottom) after right visual stimulations.