[Reviews · NeurIPS 2019]

Reviewer 1



In this paper, the authors propose CLaR (Concomitant Lasso with Repetitions) which is an approach to solve lasso problems with multiple realizations and heteroskedastic noise. Overall this is a reasonably good paper although I found parts of it to be not explained very clearly. The authors provide a theoretical justification for their approach, an algorithm to solve their problem and some applications in simulated and real MEG data. Technically, this is a reasonably sound paper. The objective function combines a novel loss function with the L 2,1 norm to promote sparsity. My main comment is that the motivation for the approach is hard to follow, because the authors introduce a lot of different concepts (heteroscedacity, concomitant estimation of coefficients and noise, and M/EEG data) without explaining well the connections between them very well. For example, why does increasing the number of observations lead to heterocedasticity? (line 26) What are the repetitions the authors are combining (e.g. in an MEEG context)? I would suggest a modest re-writing of the introduction to make this clearer. After a couple of readings it seems that the approach was specifically designed with M/EEG in mind. If that is the case, I would make that case explicitly in the introduction and please also provide a more detailed explanation of the M/EEG problem setup in the main text of the paper - currently this is not very clear. Moreover, the authors approach is also couched in multi-task learning (MTL) terminology without this connection being made fully explicit In the real data experiments, the authors set the regularization parameter on the basis of their expectation to see only two sources in the data. This may be reasonable for the data they are evaluating, but this knowledge is unlikely to be true in general. Perhaps the authors can comment on that. How would they then set the regularization parameter in other settings? Other comments: - the authors seem to abuse their notation. on line 65, the authors define $||.||_{p1}$ to mean the l_p1 norm for any $p \in [1,\inf]$ but then they use this notation to mean the norm induced by the estimated precision matrix $S^{-1}$ (e.g. in eq. 2). Is this a misunderstanding? - line 45: probably should be "reduces noise variance inversely proportional to the number of repetitions". - _why_ does the homoscedastic solver fail after whitening? (line 222)

Reviewer 2



The paper is well-written and the approach is interesting, however the paper contributions listed above highly overlap with a previous work [Massias 2018a AISTATS]. The differentiating part between two works is that the previous approach averaged noisy observations, whereas this work slightly updates the previous solver to minimize a data fidelity term which is the summation over all repetitions. It is empirically shown to have some advantages over the averaging approach (this point was also emphasised in the supplementary part B.8).

Reviewer 3



Originality: The paper seems original compared to a set of alternatives that the authors have provided and compared against. Its potential for neuroimaging applications such as M/EEG is also interesting. Quality: The paper is overall well-written. Reproducibility is much appreciated, and the supplement preemptively answered some of my questions related to the experiments. Clarity: For the most part, it was written clearly so I was able to follow the crucial points for the most part given my limited knowledge in the area. The provided code helped me a lot to digest some technical pieces of the method. However, the neuroimaging related explanations were relatively less clear than the other part which I will describe below. Significance: Given my understanding of the paper, I was able to appreciate the contribution of the method towards some real data applications. Thanks to the provided code, the impact of the paper could be immediate and more probable.

[Author Response · NeurIPS 2019]

**R1**:

- **Motivation and context:** the proposed estimator is motivated with M/EEG applications in mind. For the sake of clarity, the M/EEG part of the introduction has been rewritten thanks to the insightful comments. We now detail that in the M/EEG setting the noise is correlated between sensors (non-diagonal covariance), but also that three types of sensors are potentially available (gradiometers, magnetometers and electrodes). We now also clarified that 3 types of sensors can be used to increase the number of measurements (samples), yet each sensor type measures different physical quantities and have different noise characteristics (heteroscedasticity). To avoid any ambiguity we will replace the word 'heteroscedastic' by 'correlated' in the title. MTL has been recalled in the introduction to make the connection more explicit.

- **Repetitions in the M/EEG context:** the repetitions concern the cognitive experiment (eg, recording M/EEG signal for 1 s following an auditory stimulation on one patient). The same experiment is performed, typically 50 times, sequentially on the same patient, which results in 50 repetitions of all sensors measurements. Alternative word for repetition in this context is *trial*.

- **Hyperparameter tuning** is a difficulty shared by all the compared methods. Popular approaches include (generalized) cross-validation. We wanted to decouple the two possible causes of errors: the one due to imperfect hyperparameter setting and the one due to the estimator itself. That's why **1)** on synthetic and realistic data we provided support recovery ROC curves (with a wide range of $\lambda$s) **2)** on real data we fixed the number of non-zero coefficients to 2 (for this dataset using auditory stimuli one expects one active source in each auditory cortex), and we selected a corresponding $\lambda$. It allowed us to compare methods performance irrespective of the selection of the regularization parameter.

- **Notation:** when the norm subscript is a real number $p$, $||.||_p$ refers to the classical $\ell_p$ norm (L66), and for a (positive definite) matrix $S^{-1}$, it refers to the associated Mahalanobis norm (L71). We will detail this subtlety in the notation.

- *"why does the homoscedastic solver fail after whitening? (L222)"*: we may have been unclear. The sentence L222 is "**without** this whitening process, the homoscedastic solver MTL fails". Indeed MTL is the $\ell_{2,1}$ regularized Maximum Likelihood Estimator with an iid (white) Gaussian noise modelling assumption. We meant that when this iid assumption breaks, e.g. when data are not whitened, MTL fails. We will rephrase to avoid any confusion.

**R2**: **Overlap with Massias 2018a** Our work builds actually on Massias 2018a. Whereas this paper focused on how to solve optimization problems like SGCL (Eq. 4), our main theoretical contribution is to show how SGCL and CLaR result from the smoothing theory of Nesterov applied to a matrix valued function. As pointed out by R1: *"the authors provide a theoretical justification for their approach"* while the formulation introduced in Massias 2018a was **heuristic**.

**Theoretically** we stand from Massias 2018a by providing new contributions:

- **1-** We provide an **explicit variational formula** for the (smoothed) nuclear norm (see Proposition 4). To be more explicit: when $ZZ^\top \succ 0$, $\|Z\|_{\mathscr{S},1} = \min_{S \in \mathcal{S}_{++}^n} \frac{1}{2} \|Z\|_{S^{-1}}^2 + \frac{1}{2}\operatorname{Tr}(S)$ (see van de Geer 2016, Lemma 3.4, p. 37). When $ZZ^\top \not\succ 0$, one can approximate $\|Z\|_{\mathscr{S},1}$ by the following formula: $\|Z\|_{\mathscr{S},1} \approx \min_{S \in \mathcal{S}_{++}^n} \frac{1}{2} \|Z\|_{S^{-1}}^2 + \frac{1}{2}\operatorname{Tr}(S) + \frac{1}{2}(\underline{\sigma}/2)^2 \operatorname{Tr}(S^{-1}) = \sum_i \sqrt{\gamma_i^2 + (\underline{\sigma}/2)^2}$ (see Optimization with Sparsity-Inducing Penalties, Bach et al. 2012, p. 62), which is a $\underline{\sigma}/2$-smooth $\underline{\sigma}(n \wedge q)/2$-approximation of $\sum_i \gamma_i = \|Z\|_{\mathscr{S},1}$ (see Beck and Teboulle 2012, Example 4.6, p.573). Here, we proposed a different smoothing $\left(\|\cdot\|_{\mathscr{S},1} \square \omega_{\underline{\sigma}}\right)$ for which we provide an **explicit formula**: $\left(\|\cdot\|_{\mathscr{S},1} \square \omega_{\underline{\sigma}}\right)(Z) = \min_{S \succeq \underline{\sigma}\operatorname{Id}} \frac{1}{2} \|Z\|_{S^{-1}}^2 + \frac{1}{2}\operatorname{Tr}(S)$ (Prop. 4) which is a $\underline{\sigma}$-smooth $\underline{\sigma}(n \wedge q)/2$-approximation (see Beck and Teboulle 2012, Thm. 4.1, p. 567) of $\|\cdot\|_{\mathscr{S},1}$. This smooth approximation leads to a better Lipschitz constant for a given $\varepsilon$-approximation. We will highlight such a **theoretical** and practical **benefit** (faster convergence thanks to a better Lipschitz constant) of our approach.

- **2-** The proposed smoothing approach (see App. A) paves the way to a practical use of Schatten norms as datafitting terms by not requiring to solve problems where both datafitting and regularization terms are non-smooth; see in particular the smoothing of the Schatten 2 (App. A.5) and Schatten $\infty$ (App. A.6) -norm.

**Empirically** we stand from Massias 2018a with the following contributions: **1)** the **modelling** contribution with the repetitions for M/EEG **2)** the **extensive benchmark** against convex and non-convex estimators (with a clean opensource package pointed out by R3 "the provided code helped me a lot to digest some technical pieces") **3)** **extensive experiments** on **real data** (Fig.6 and 7 + App. D) with potential impact for the neuroscience community (R3 "Thanks to the provided code, the impact of the paper could be immediate and more probable"). **4)** We report that solving CLaR is as computationally cheap as solving SGCL, see App. B.7 and Tab. 1.

**R3**:

- **clarifications about M/EEG context**: (see answer to R1) we will better explain the specificities of the M/EEG framework in the introduction.

- *sample* is a publicly available M/EEG dataset included in the Python package MNE. It consists in measurements $Y^{(1)}, \ldots, Y^{(r)}$ corresponding to auditory or visual stimulations. This has been clarified in the paper.

[Meta-Review · NeurIPS 2019]

The paper provides the formulation and the code for an efficient algorithm to solve lasso problems with multiple realizations and heteroskedastic noise. This method has a great potential impact on M/EEG data analysis. As noted by one Reviewer since the contribution is closely related to a previous work the paper migth be considered an incremental work more suitable for a poster presentation.